# A Swiss Army Knife for Heterogeneous Federated Learning: Flexible Coupling via Trace Norm

**Tianchi Liao, Lele Fu, Jialong Chen, Zhen Wang, Zibin Zheng, Chuan Chen***
Sun Yat-sen University, Guangzhou, China
{liaotch, fulle, chenjlong}@mail2.sysu.edu.cn
{wangzh665, zhzibin, chenchuan}@mail.sysu.edu.cn

## Abstract

The heterogeneity issue in federated learning (FL) has attracted increasing attention, which is attempted to be addressed by most existing methods. Currently, due to systems and objectives heterogeneity, enabling clients to hold models of different architectures and tasks of different demands has become an important direction in FL. Most existing FL methods are based on the homogeneity assumption, namely, different clients have the same architectural models with the same tasks, which are unable to handle complex and multivariate data and tasks. To flexibly address these heterogeneity limitations, we propose a novel federated multi-task learning framework with the help of tensor trace norm, FedSAK. Specifically, it treats each client as a task and splits the local model into a feature extractor and a prediction head. Clients can flexibly choose shared structures based on heterogeneous situations and upload them to the server, which learns correlations among client models by mining model low-rank structures through tensor trace norm. Furthermore, we derive convergence and generalization bounds under non-convex settings. Evaluated on 6 real-world datasets compared to 13 advanced FL models, FedSAK demonstrates superior performance.

## 1 Introduction

Federated learning (FL) is an effective machine learning approach that enables decentralized computations and data resources [1]. It is regarded as a promising distributed and privacy-preserving method. A common challenge in FL is the data heterogeneity problem, in particular when the diversity of client data distribution increases, the generalization error of the global model increases significantly as well [2]. Therefore, to address data heterogeneity, common personalized federated learning (pFL) methods learn a personalized model for each client in addition to the global model [3, 4, 5].

However, current research indicates that the fundamental bottleneck in executing pFL across heterogeneous clients is the misassumption of one global model can fit all clients [6]. Instead, we should focus on exploring intrinsic collaborations across clients to obtain better local models. Unlike pFL, the goal of federated multi-task learning (FMTL) is to simultaneously learn separate models, where each model caters to the heterogeneous needs of each client [7]. Thus, FMTL directly addresses the issues stemming from client heterogeneity without constructing a global model [8].

Moreover, most of the existing FL methods require all clients to train models with the same architecture (i.e., model homogeneity) [9, 10, 11]. In practical heterogeneous FL scenarios, besides data heterogeneity, model heterogeneity and task heterogeneity are also present due to varying hardware, computational capabilities, and requirements across clients [12, 13]. Although some FL methods to handle model heterogeneity have emerged [14, 15, 16], many of them resort to knowledge distillation

---

*Corresponding author

38th Conference on Neural Information Processing Systems (NeurIPS 2024).

techniques that necessitate public datasets closely aligned with the learning objectives [17, 18]. This incurs high communications and computational costs, limiting model performance. Alternatively, some strategies utilize prototype models linked to labels [15, 16], rendering them futile when client tasks are related but inconsistent.

Currently, task heterogeneity is rarely mentioned in federated settings, however, it is widespread in real-world scenarios [19]. For example, given the same batch of portrait samples for different clients, some clients may want to predict person's age (Task 1: Multi-Class Classification or Regression), while others may need to recognize gender (Task 2: Binary Classification). In such scenario, existing methods are ill-equipped to handle task heterogeneity. Effective algorithms that can overcome data, model, and task heterogeneity in federated settings remain largely underdeveloped.

In light of the heterogeneity in FL and limitations of existing techniques, we adopt FMTL to address complex heterogeneous FL. A key issue in FMTL is how to design priors such that knowledge obtained from each client can be shared by others. Corinzia *et al.* [20] built connections among client tasks using approximate variational inference, while Dinh *et al.* [8] proposed a Laplace regularization-based FMTL that only considers grouping similarities among different client tasks. To effectively utilize correlations across clients, Kumar *et al.* [21] proposed a method leveraging low-dimensional subspaces shared by multiple tasks, which was shown effective but limited to linear models.

Imposing low-rank constraints on model parameters is effective when learning objectives are correlated among clients [22]. Trace norm has been proposed as a solution to uncover potential connections among model parameters of different objectives. Thus, we propose a novel and flexible FMTL framework based on tensor trace norm, **FedSAK**, which like a **S**wiss **A**rmy **K**nife provides flexible aggregation choices for heterogeneous FL. Specifically, we split each client model into a feature extractor and a prediction head, allowing our model can adaptively define certain parts as global shared layers for different heterogeneity settings and upload them to the server. The server aggregates the global shared layers into a tensor and applies trace norm regularization to induce a low-rank structure. In this way, inter-dependencies are created among different client models to reflect across clients' intrinsic connections about their model parameters. In summary, our main contributions are:

- FedSAK is an FMTL algorithm that simultaneously considers data heterogeneity, model heterogeneity and task heterogeneity. It is more flexible than most existing FL methods.

- We employ tensor trace norm to exploit low-rank structure for identifying relationships among client models.

- We theoretically derive convergence guarantees for FedSAK under non-convex settings, and establish generalization bounds for the proposed tensor trace norm minimizer.

- We conduct extensive experiments on 13 advanced methods over 6 datasets to demonstrate the flexibility and efficacy of FedSAK. Results show that FedSAK outperforms baselines in handling heterogeneous federated scenarios.

## 2 Related Works

### 2.1 Heterogeneous Federated Learning

**Data Heterogeneity**, is one of the most significant challenges in FL [23]. Initial methods like FedProx [2] added a proximal term to the local training objective to keep updated parameters close to the original downloaded model. MOON [9] employed contrastive loss to improve representation learning. Additionally, various personalized models have been proposed to train specialized components using globally shared information, including fine-tuning methods like FedRep [24], FedPer [25]; regularization-based methods such as FedMTL [7], pFedMe [4], and Ditto [5]; meta-learning methods like Per-FedAvg [3]; and methods decoupling feature extractors and classifiers like GPFL [10], FedCP [26]. Moreover, the clustered FL has also been explored by partitioning clients into multiple groups or clusters for clustered local models to provide multiple global models [27]. However, the development of existing data heterogeneity methods is constrained by homogenous model assumptions.

**Model Heterogeneity** presents another major challenge in FL. Researchers often employ FL based on knowledge distillation as an alternative solution. FedMD [17] have clients compute logits on a public dataset using locally trained heterogeneous models, which are then uploaded to the server. FedDF [28] and FedKT [29] train each client's heterogeneous model on a shared public dataset at the

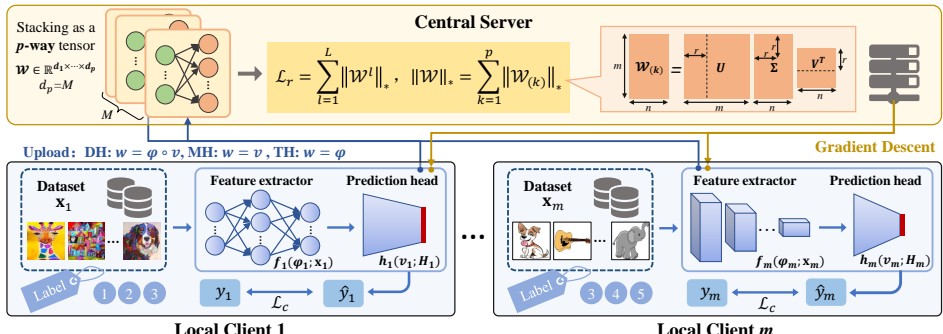

Figure 1: The main framework of FedSAK model. DH denotes "Data Heterogeneity", MH denotes "Model Heterogeneity", and TH denotes "Task Heterogeneity".

server via distillation. However, obtaining shared datasets with similar data distributions required by these methods may not always be feasible in a practical setting. Especially when the public dataset is large-scale, the computational costs of such methods can increase substantially, limiting their applicability. Additionally, there are FL methods of model heterogeneous that employ aggregated logit or representation matching losses by class to train local models e.g., FedProto [15] and FedGH [16]. However, these methods impose higher computational costs on clients. In addition, each client can only acquire knowledge of known categories from the server, restricting generalization to unseen categories as well as possibilities for task heterogeneity.

**Task Heterogeneity** is an often overlooked issue in federated settings where tasks may have varying numbers of outputs in practice. Huang *et al.* [30] proposed a model for multi-lingual speech processing with each task corresponding to an individual language. Zhang *et al.* [31] employed DNNs for facial landmark localization and face attribute recognition. However, these task-heterogeneous methods have not been considered for FL. Yao *et al.* [32] first generalized traditional FL to federated heterogeneous task learning, but they did not propose a new algorithm and merely considered data and task heterogeneity.

## 2.2 Federated Multi-Task Learning

Multi-task learning (MTL) aims to process all task data with identical distributions centrally in a single computing environment [33]. In contrast, FMTL places greater emphasis on data privacy and heterogeneity. FMTL aims to learn separate models tailored to local heterogeneous conditions for each client. FMTL was first introduced in [7], where a system-aware optimization method was proposed that first considered theoretical issues like high communication costs, stragglers, and fault tolerance in FMTL. Yasmin *et al.* [34] formulated FMTL network using generalized total variation minimization as a regularizer. Li *et al.* [35] adopted FMTL algorithms to handle accuracy, fairness and robustness issues in FL. Corinzia *et al.* [20] modeled the FL network as a star-shaped Bayesian network and used approximate variational inference for FMTL. Dinh *et al.* [8] utilized Laplace regularization to construct relationships among clients. However, these methods operate under model homogeneity assumptions, and there is scarce analysis of the convergence and error bounds for non-convex FMTL objectives. To effectively utilize correlations across tasks, some MTL methods such as Maurer *et al.* [33] established excess risk bounds for MTL based on data distributions. Kumar *et al.* [21] proposed a new framework where in-group tasks lie in a low-dimensional subspace. Zhang *et al.* [22] employed transformed tensor nuclear norm constraints to capture intrinsic relationships among tasks. However, these methods did not explore federated settings and heterogeneity.

## 3 Notations and Preliminaries

### 3.1 General Federated Multi-Task Learning

Suppose we have $M$ clients, where client $i$ has $n_i$ private data points $\mathbf{x}_i \in \mathbb{R}^{d_{\mathbf{x}} \times n_i}$ with labels $y_i$, and $N$ denotes the total number of data, i.e., $N = \sum_{i=1}^{M} n_i$. The datasets among the clients are heterogeneous. With the help of a central server, the goal of FMTL is to collaboratively learn

individual local models $\theta_i \in \mathbb{R}^{d_\mathbf{x} \times d_\theta}$ for each client's data without exchanging private data. Note that when the model is a shallow network, $d_\theta = 1$. Many FMTL problems can be captured by the following general formulation [7]:

$$\min_{\Theta, \Omega} \left\{ \sum_{i=1}^{M} \sum_{j=1}^{n_i} \mathcal{L}_i \left( \mathcal{F}_i(\theta_i; \mathbf{x}_i^j), y_i^j \right) + R(\Theta, \Omega) \right\}, \tag{1}$$

where $\Theta := [\theta_1, \cdots, \theta_m] \in \mathbb{R}^{d_\mathbf{x} \times d_\theta \times M}$ is a collective model stacked by individual clients. $R(\cdot)$ is a regularizer, and $\Omega$ is expressed as modeling the relationship among client tasks. FMTL issues vary according to their presuppositions about $R$, which receives $\Omega$ as input and promotes some suitable structure amongst the tasks.

### 3.2 Tensor Trace Norm

In the field of MTL, the trace norm is utilized as a regularization method to learn the low-rank structure among model parameters across all tasks [22]. Typically, when dealing with vectorized data, shallow networks are used, i.e., $d_\theta = 1$, where $\Theta$ is a matrix. Thus, matrix trace norm is employed to enhance dependencies among models, defined as $\|\Theta\|_* = \sum_i \sigma_i(\Theta)$, where $\sigma_i(\Theta)$ denotes the $i$-th largest singular value of a matrix. However, with the collection of complex data, the data can be a tensor (e.g., images) and the model can become more complex (e.g., deep neural networks). In such scenarios, the parameters for all tasks can be structured into a multi-dimensional tensor, such as a $p$-way tensor ($p \geq 3$), e.g., $\Theta \in \mathbb{R}^{d_1 \times \cdots \times d_p}$. For example, in a classification model with a fully connected layer (i.e., $p = 3$), where $d_1$ represents the dimension of the input, $d_2$ indicating the number of classes, and $d_3 = M$ denotes the number of clients. In this context, the traditional matrix trace norm becomes inapplicable, necessitating the use of a tensor trace norm instead.

Currently, the Tucker trace norm, a representative overlapping tensor trace norm, is extensively utilized in deep learning [36]. It unfolds the tensor into a matrix using Tucker decomposition and computes the convex sum for the matrix trace norms of the various flattened tensors [37]. The "unfold" operation along the $k$-th mode on a tensor $\Theta$ is defined as $\mathrm{unfold}_k(\Theta) := \Theta_{(k)} \in \mathbb{R}^{d_k \times (d_1 \cdots d_{k-1} d_{k+1} \cdots d_p)}$. $\alpha_k$ denotes the weight of the matrix unfolded along $k$-th mode. Thus, we formulate the Tucker-based trace norm in the following form:

$$\|\Theta\|_* := \sum_{k=1}^{p} \alpha_k \left\| \Theta_{(k)} \right\|_*, \tag{2}$$

where $\alpha_k$ is the weight on the $k$-th mode, here we default to the same weight on each mode. Thus, the computational complexity of the tensor trace norm is $O(\min_k d_k^2 \prod_{i \neq k}^{p} d_i)$.

## 4 Methodology

In this section, we introduce the multi-task learning framework in a federated environment in more detail and propose a novel method, FedSAK, to address the challenges of FMTL. The framework of FedSAK is displayed in Figure 1.

### 4.1 Optimization Objective

Without loss of generality, the client model can be decoupled into representation layers, also known as a feature extractor, and final decision layers like a prediction head for classification tasks. Under this design, much research has actively studied collaboration among different layers. However, these methods typically require model homogeneity as a means for server parameter aggregation, which inherently restricts the development of heterogeneous FL. Therefore, our objective in FMTL is to facilitate heterogeneous FL in supervised classification scenarios, encompassing both data heterogeneity (DH), model heterogeneity (MH), and task heterogeneity (TH).

Following previous conventions, we denote the feature extractor as $f_i(\varphi_i; \mathbf{x}_i) : \mathbb{R}^{d_\mathbf{x}} \to \mathbb{R}^{d_H}$ and the prediction head as $h_i(v_i; c_i) : \mathbb{R}^{d_H} \to \mathbb{R}^{d_y}$. Thus, the model for client $i$ can be expressed as $\mathcal{F}_i(\theta_i) = f_i(\varphi_i) \circ h_i(v_i)$, $\circ$ denotes concatenation among model components, where $\theta_i$ are the model parameters. We assume $f_i$ and $h_i$ can be heterogeneous across clients, meaning clients can

customize the size and architecture of their local feature extractor and prediction head based on available resources.

To flexibly apply our method to different federated heterogeneity settings, we define $w_i$ to represent the global shared layers for client $i$, which is a subset of $\theta_i$, i.e. $w_i \subseteq \theta_i$. The choice of $w_i$ can be flexibly adapted to the heterogeneity setting. For example, with data heterogeneity, $w_i = \theta_i = \varphi_i \circ v_i$; with model heterogeneity, $w_i = v_i$; and with task heterogeneity, $w_i = \varphi_i$.

Since the global shared layers are aggregated at the server, where a low-rank structure among clients is learned by computing the trace norm to reinforce dependencies among models. Thus the objective for heterogeneous FMTL in Eq. (1) can be reformulated as:

$$\min_{\Theta} \frac{1}{M} \sum_{i=1}^{M} \frac{1}{n_i} \sum_{j=1}^{n_i} \mathcal{L}(\mathcal{F}_i(\Theta; \mathbf{x}_i^j), y_i^j) + \lambda \left\| \mathcal{W} \right\|_*, \tag{3}$$

where $\mathcal{F}_i(\theta_i) = f(\varphi_i) \circ h(v_i)$, we use $\theta_i$ to represent $(\varphi_i, v_i)$ for short, and $\Theta = [\theta_1, \cdots, \theta_M]$. $\mathcal{W} \in \mathbb{R}^{d_1 \times \cdots \times d_p}$ ($d_p = M$) is denoted as the tensor stacked by each client's shared layers $w_i$ and $\lambda$ is the hyperparameter.

## 4.2 Global Shared Layer Representation

Considering the heterogeneity of participating clients, the optimal model parameters are not identical across clients. This implies that simple weighted aggregation is insufficient to provide the required information to each client. Therefore, to uncover the underlying connections among model parameters of different clients, we utilize the trace norm as a regularizer to induce a low-rank structure among the parameters, which can better exploit intrinsic task relationships among the clients in the FMTL fashion. Since deep learning has evolved, the global shared layers $w$ may be an $L$-layer deep network structure, we use the superscript $l$ to denote the $l$-th layer of the global shared layers. Specifically, we first stack the global shared layers into a tensor on the server, which can handle the inherent correlation among multiple local models more efficiently:

$$\mathcal{W}^l = \text{stack}(w_1^l; w_2^l; \cdots; w_M^l) \in \mathbb{R}^{d_1 \times \cdots \times d_p}, \tag{4}$$

where $p$ denotes that $\mathcal{W}^l$ is a $p$-way tensor, $d_1$ to $d_{p-1}$ are denoted as the dimensions of the model parameters and $d_p$ is the number of clients, i.e., $d_p = M$. Then, we regularize the tensor $\mathcal{W}^l$ formed by stacking the global shared layers $w_i^l$ from all $M$ clients using a trace norm penalty $|| \cdot ||_*$. That is:

$$\mathcal{L}_r = \sum_{l=1}^{L} \left\| \mathcal{W}^l \right\|_* = \sum_{l=1}^{L} \sum_{k=1}^{p} \left\| \mathcal{W}_{(k)}^l \right\|_*, \tag{5}$$

where $\mathcal{W}_{(k)}^l$ is the matrix unfolded according to the $k$-th dimension, see Eq. (2). Minimizing the trace norm of $\mathcal{L}_r$ yields a low-rank structure that reveals commonalities among the clients. This allows clients to transfer knowledge through coupled shared low dimensional subspaces, while still allowing clients to customize local models $f_i$ and $h_i$.

For the server's $t$-th round of global shared layers training, after stacking the received global shared layers form clients to compute the trace norm loss, we update the global shared parameters $\widetilde{w}_i^t$ by gradient descent. Since the trace norm of a matrix is not differentiable, according to Watson *et al.* [38], we can compute a subgradient,

$$\frac{\partial \left\| \mathcal{W}_{(k)}^l \right\|_*}{\partial \mathcal{W}_{(k)}^l} = UV^T, \tag{6}$$

where for $\mathcal{W}_{(k)}^l = U\Sigma V^T$, the singular value decomposition of $\mathcal{W}_{(k)}^l$. Since $w_i$ is the $i$-th slice of $\mathcal{W}$, for each client we can update the global shared layers in a slice-wise manner as:

$$\widetilde{w}_i^t \leftarrow w_i^t - \eta_w \nabla \mathcal{L}_r, \tag{7}$$

where $\eta_w$ is the learning rate. Intuitively, this subtracts the aggregated trace norm subgradient from each client's current shared layers, reducing redundancy and coupling the parameters to learn a jointly low-dimensional subspace. The updated $\widetilde{w}_i^t$ are then sent back to the respective client at the end of each communication round to update their local models for the next round of training.

## 4.3 Local Model Update

The server broadcasts the updated global shared layers $\widetilde{w}_i^t$ to the client. In the $(t+1)$-th round, client $i$ replaces the shared layers of its local model with the received global shared layers. Thus, the replaced local model is represented as $\widetilde{\theta}_i^{t+1}$.

Each client's local model obtains local knowledge from the local update and further obtains global knowledge among clients from the global update, allowing it to better handle heterogeneity. The assembled full local model $\theta_i$ is then trained on the local data $\mathbf{x}_i$ to obtain the updated local model parameters:

$$\theta_i^{t+1} \leftarrow \widetilde{\theta}_i^{t+1} - \eta_\theta \nabla \mathcal{L}_c \left( \widetilde{\theta}_i^{t+1}; \mathbf{x}_i \right), \tag{8}$$

where $\eta_\theta$ is the learning rate, and $\mathcal{L}_c$ is the loss of cross entropy. We summarize the steps of FedSAK in Algorithm 1, see Appendix.

## 5 THEORETICAL ANALYSIS

### 5.1 Convergence Analysis

To analyze the convergence of FedSAK, we define $t$ as the current communication round, $e \in \{0, 1, \cdots, E\}$ as the number of local iterations, where $E$ denotes the maximum number of local iterations. Thus, $(tE + e)$ represents the $e$-th iteration in the $(t+1)$-th communication round. The $(t+0)$ denotes that at the beginning of the $(t+1)$-th round, the client uses the global shared layers gradients from round $t$ to update the local shared layer parameters. Note that $(tE + E)$ corresponds to the last iteration in round $(t+1)$. We make some assumptions see Appendix C.1, which is similar to the existing general framework [15, 16]. Based on the above assumptions, due to our same local training, Tan *et al.* [15] and Yi *et al.* [16] deduce that Lemma 1 and 2 still holds. For notational simplicity, we set $\eta = \eta_\theta = \eta_w$.

**Lemma 1** *Based on Assumption 1 and 2, in the local iteration $e \in \{0, 1, ..., E\}$ of the $(t+1)$-th training round, the local model loss of any client is bounded by.*

$$\mathbb{E}\left[ \mathcal{L}_{(t+1)E} \right] \leqslant \mathcal{L}_{tE+0} - \left( \eta - \frac{K\eta^2}{2} \right) \sum_{e=0}^{E} \| \mathcal{L}_{tE+e} \|_2^2 + \frac{KE\eta^2}{2}\sigma^2. \tag{9}$$

**Lemma 2** *Based on Assumption 3, the loss of an arbitrary client's local model $(t+1)$-this bounded by:*

$$\mathbb{E}\left[ \mathcal{L}_{(t+1)E+0} \right] \leqslant \mathbb{E}\left[ \mathcal{L}_{(t+1)E} \right] + \frac{\eta K \lambda \omega^2}{2}. \tag{10}$$

The detailed proof can be found in Appendix C.2-C.3 [15, 16].

Based on Lemma 1 and Lemma 2, we can derive the model nonconvex convergence rate.

**Theorem 1** *The above assumptions, for an arbitrary client and any $\epsilon > 0$, if $\eta < \frac{2E\epsilon - K\lambda\omega^2}{KE(\epsilon + \sigma^2)}$, the following inequality holds:*

$$\frac{1}{TE} \sum_{t=0}^{T-1} \sum_{e=0}^{E} \mathbb{E}\left[ \| \mathcal{L}_{tE+e} \|_2^2 \right] \leqslant \frac{2\left( \mathcal{L}_{t=0} - \mathcal{L}^* \right)}{TE\eta\left( 2 - K\eta \right)} + \frac{K\left( E\eta\sigma^2 + \lambda\omega^2 \right)}{E(2 - K\eta)} \leqslant \epsilon, \tag{11}$$

With this, it is evident that the local model of any client of FedSAK converges at a non-convex convergence rate $\mathcal{O}\left( \frac{1}{T} \right)$. See Appendix C.4 for a detailed proof.

### 5.2 Excess Risk Bound

Without loss of generality [37, 39], to simplify the analysis we take the case where $\Theta = \mathcal{W}$, i.e., sharing all model structures. Consequently, we define the problem (3) empirical loss for all the tasks as

$$\min_{\mathcal{W}} \hat{\mathcal{R}}(\mathcal{W}) = \sum_{i=1}^{M} \frac{1}{n_i} \sum_{j=1}^{n_i} \mathcal{L}(\mathcal{F}_i(\mathcal{W}; \mathbf{x}_i^j), y_i^j) \quad \text{s.t. } \|\mathcal{W}\|_* \leq \gamma. \tag{12}$$

where $\gamma$ is a regularization parameter that controls the complexity of the model. Follow [37], the true risk is defined by the generalized loss of the task as $\mathcal{R}(\mathcal{W}) = \sum_{i=1}^{M} \mathbb{E}_{(\mathbf{x},y) \sim \mathcal{P}_i} \mathcal{L}(\mathcal{F}_i(\mathcal{W}; \mathbf{x}), y)$, where $\mathcal{P}_i$ denotes the underlying data distribution for the $i$-th client and $\mathbb{E}[\cdot]$ denotes the expectation. Each training data $x_j^i$ is assumed to satisfy $\langle x_j^i, x_j^i \rangle \leq 1$. To characterize correlations among features, we assume that $\mathbf{C}_k = \mathbb{E}[\mathbf{x}_{(k)}^{i,j}(\mathbf{x}_{(k)}^{i,j})^T] \preceq \frac{k}{d}\mathbf{I}$ for any $k \neq \emptyset$ and $k \subseteq [p-1]$, where $\mathbf{A} \preceq \mathbf{B}$ means that $\mathbf{B} - \mathbf{A}$ is a positive semidefinite matrix, $d = \prod_{i \in [p-1]} d_i$, and $\mathbf{I}$ denotes an identity matrix with an appropriate size.

**Lemma 3** *From Tomioka et al. [40], the dual norm of the tensor trace norm in Eq. (2) is defined as*

$$\|\mathcal{W}\|_{*\star} = \inf_{\sum_{k \neq \emptyset, k \subset [p]} \mathcal{W}^{(k)} = \mathcal{W}} \max_k \left\|\mathcal{W}_{(k)}^{(k)}\right\|_{\infty}, \tag{13}$$

*where the infimum is over tensors $\mathcal{W}^{(k)}$ that sum to the original tensor $\mathcal{W}$, and $\|\cdot\|_{\infty}$ is the operator norm (maximal singular value).*

For simplicity, different tasks are assumed to have the same number of data points, i.e., $n_i$ equals $n$ for $i = 1, \cdots, M$. It is simple to extend our analysis to the general case.

**Theorem 2** *Let $\sigma_i^j$ be a Rademacher variable, which is a random variable taking values in $\{\pm 1\}$ with equal probability. Consider $\mathcal{W}$ to be a tensor of order $d_1 \times \cdots \times d_{p-1} \times d_p$ with $\mathcal{W}_i = \sum_{j=1}^{n} \frac{1}{n}\sigma_i^j \boldsymbol{x}_i^j$, where $d_p$ is set equal to M. Then the following inequality holds:*

$$\mathbb{E}[\|\mathcal{W}\|_{*\star}] \leq C \min_k \left( \sqrt{\frac{\kappa M}{nd} ln \, D_k} + \frac{ln \, D_k}{n} \right), \tag{14}$$

*where $D_k = d_k + (d_1 \cdots d_{k-1} d_{k+1} \cdots d_p)$, and $C, \kappa$ are an absolute constant.*

Under Assumption 1 and Theorem 2 we derive the excess risk bound of the estimator in Eq. (12).

**Theorem 3** *Suppose that $n_i \geq n > 0$, $|y_i^j| \leq b$, then for any $\|\mathcal{W}\|_* \leq \gamma$ and $\delta \in (0, 1)$, the following inequality holds*

$$\mathcal{R}(\hat{\mathcal{W}}) - \mathcal{R}(\mathcal{W}) \leq \frac{4\gamma K}{M}\mathbb{E}[\|\mathcal{W}\|_{*\star}] + \frac{2bK\sqrt{N}}{Mn} + a\sqrt{\frac{2log(2/\delta)}{Mn}}, \tag{15}$$

*with probability at least $1 - \delta$, where $\hat{\mathcal{W}}$ is the optimal solution of Eq. (12) and $\|\mathcal{W}\|_{*\star}$ is defined as Theorem 2.*

Table 1: Test accuracy (%) on image classification tasks under data heterogeneity.

| Model | HumA | | MNIST | | CIFAR-10 | | | | CIFAR-100 | | | | Promote |
|---|---|---|---|---|---|---|---|---|---|---|---|---|---|
| (# $M$, # $S$) | (10,6) | (30,6) | (20,2) | (100,2) | (10,3) | (10,5) | (20,3) | (20,5) | (10,30) | (20,10) | (20,30) | (20,50) | $\Delta$ |
| FedAvg | 93.71 | 95.65 | 82.69 | 92.85 | 61.53 | 65.77 | 62.7 | 66.55 | 25.29 | 17.29 | 27.44 | 29.48 | — |
| FedProx | 91.68 | 95.28 | 88.15 | 93.72 | 61.81 | 55.02 | 52.57 | 53.77 | 23.06 | 15.89 | 23.41 | 22.7 | -3.66 |
| MOON | 91.58 | 93.65 | 87.31 | 93.35 | 64.94 | 65.41 | 59.97 | 65.7 | 25.04 | 18.71 | 29.28 | 28.51 | +0.21 |
| Per-FedAvg | 93.34 | 93.27 | 90.43 | 92.36 | 74.19 | 71.87 | 79.9 | 72.4 | 32.94 | 42.65 | 37.22 | 30.82 | +7.62 |
| pFedMe | 91.54 | 95.72 | 90.55 | 93.73 | 80.19 | 73.72 | 81.72 | 74.69 | 35.81 | 54.34 | 38.48 | 32.18 | +10.16 |
| Ditto | 92.06 | 97.63 | 96.98 | 98.13 | 80.33 | 74.93 | 81.14 | 75.58 | 34.95 | 52.62 | 35.99 | **32.97** | +11.03 |
| GPFL | 91.25 | 94.98 | 94.82 | 97.94 | 73.85 | 70.43 | 72.68 | 70.56 | 32.98 | 47.68 | 32.49 | 25.29 | +7.05 |
| FedAvgDBE | 93.96 | 95.9 | 91.85 | 95.24 | 74.12 | 70.17 | 73.01 | 71.06 | 26.46 | 35.23 | 29.64 | 29.35 | +5.42 |
| FedMTL | 94.11 | 98.15 | 98.18 | 98.66 | 78.25 | 70.36 | 79.69 | 70.96 | 35.51 | 53.62 | 34.95 | 27.74 | +9.94 |
| FedU | 92.46 | 95.86 | 95.65 | 96.95 | 77.26 | 72.47 | 81.98 | 73.98 | 35.73 | 52.26 | 36.27 | 31.11 | +10.17 |
| FedMD | 90.46 | 96.51 | 89.54 | 92.54 | 75.65 | 68.64 | 80.28 | 71.03 | 29.66 | 50.43 | 30.58 | 28.39 | +6.90 |
| FedProto | 97.71 | 98.03 | 98.04 | 98.24 | 83.05 | 72.11 | 82.13 | 75.83 | 35.85 | 55.04 | 37.73 | 30.9 | +11.98 |
| FedGH | 91.23 | 98.45 | 98.33 | 98.28 | 82.17 | 72.18 | 79.97 | 72.69 | 34.77 | 52.3 | 34.91 | 25.22 | +9.96 |
| FedSAK | **98.46** | **99.28** | **98.58** | **98.85** | **83.71** | **75.89** | **84.49** | **76.47** | **36.97** | **55.75** | **39.16** | 31.47 | +13.2 |

# 6 Experiments

## 6.1 Experimental Setup

**Datasets and local models.** We evaluate FedSAK on diverse image classification tasks. For image classification, six well-known datasets are utilized, including Human Activity (HumA) [41],

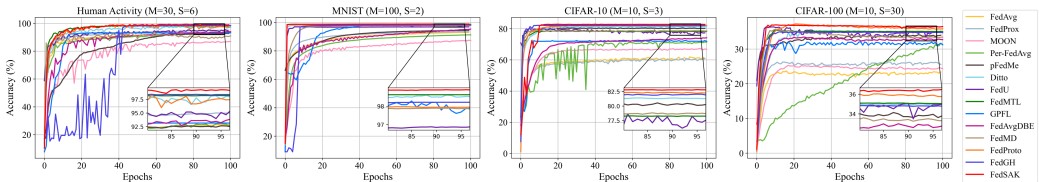

Figure 2: The test accuracy and convergence process of each method.

MNIST [42], CIFAR-10 [43], CIFAR-100 [43], PACS [44], and Adience Faces [45]. Due to space constraints, detailed introductions of the datasets and the experimental setup are provided in Appendix B, while their splits will be elaborated in Section 6.2 for each heterogeneous task. Each client test set follows a similar distribution to the training set. Two fully connected layers are provided for Human Activity and MNIST datasets. CIFAR-10, CIFAR-100, PACS, and Adience Faces datasets are downsampled to 32×32 size and consider a CNN model comprising 2 convolutional layers followed by 2 fully connected layers. Note that we only consider the last fully-connected layer of the model as the **predictor head**, and the other layers form the **feature extractor**. We ran each model 5 times and recorded its average value recorded as the result. Our code is available at: https://github.com/Tiaerc/FedSAK.

**Baselines.** We study the performance of FedSAK in heterogeneous settings and compare against baselines including: **(1)** *Conventional federated learning:* FedAvg [1], FedProx [2]; **(2)** *Personalized federated learning:* Per-FedAvg [3], pFedMe [4], MOON [9], Ditto [5], GPFL [10], FedAvgDBE [46]; **(3)** *Federated multi-task learning:* FedMTL [7], FedU [8] and **(4)** *Heterogeneous federated learning:* FedMD [17], FedProto [15], FedGH [16].

## 6.2 Results and Discussion

**Data Heterogeneity**. Following the FMTL work [7, 8], we adopt a commonly used setup called pathology Non-IID to simulate DH in the form of label distribution shift in FL. We partition the dataset into $M$ clients, with each client sampling data from $S$ classes, where the number of samples per class varies significantly. The distribution of client data for each dataset is shown in Appendix B.2. For extensive comparisons of the advanced baselines, we design the individual client models to be homogeneous in DH scenarios. Thus, the global shared layer for each client is the client's entire model, i.e., $w_i = \varphi_i \circ v_i$. Table 1 reports the average test accuracy across all clients, and Figure 2 summarizes the convergence behavior and performance of all methods. It can be observed that FedSAK achieves the highest accuracy in most cases, indicating that constraining through tensor trace norm facilitates transferring useful information across multiple clients, thereby improving model performance on each client. Notably, in this scenario, the lower the number of clients, the fewer the samples involved in training, and hence the accuracy is higher when the number of clients is higher, which is consistent with the results derived from our Theorem 3.

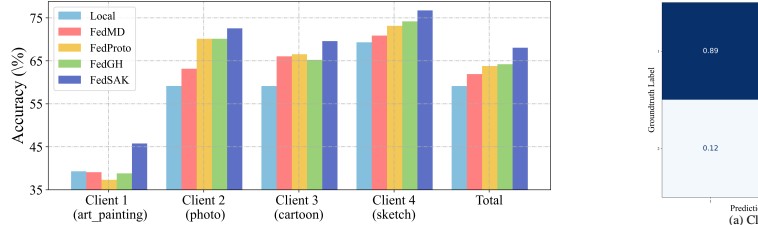
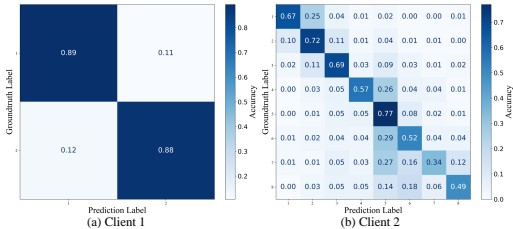

Figure 3: The test accuracy of all methods on MH. Figure 4: Confusion matrix of classification results obtained from two clients TH on the Adience Face.

**Model Heterogeneity** is an important challenge in FL due to the differences in client computational resources and the fact that heterogeneous client requires different models. For example, the dataset PACS contains 4 different domains, and we set up each client to contain data from one domain. In this setup, we vary the number of filters in the convolutional layers and dimensions of the fully connected layers to obtain 4 heterogeneous models. The detailed model architectures are shown in Table 4 in

the Appendix. Thus, in the MH scenario, our shared layers consist of the prediction head, which is $w_i = v_i$. For CIFAR10, we set up 20 clients and alter data distribution using a Dirichlet distribution and label distribution skew. PACS itself has heterogeneity, so we set 4, 8, 10 clients respectively. The results in Table 2 show that FedSAK consistently achieves the highest model accuracy, while FedMD has lower accuracy. On CIFAR-10, since the distribution of our test set is similar to that of the training set, the model's performance will decrease due to the increase in the number of predicted label categories. Therefore, the results on the Dirichlet distribution are inferior to those with skewed label distribution. On PACS, it outperforms the best baseline FedGH by 5.98%, 4.08%, and 3.94% respectively. Figure 3 shows the test set accuracy with 4 clients on this dataset. Additionally, FedSAK requires less communication, thereby converging faster.

Table 2: Test accuracy (%) for the image classification task under model heterogeneity, $\beta$ is the Dirichlet distribution, $S$ is the number of labels, and $M$ is the number of clients.

| | | Local | FedMD | FedProto | FedGH | FedSAK |
|---|---|---|---|---|---|---|
| **CIFAR-10** $M$=20 | $\beta$=0.3 | 30.67 | 32.56 | 34.15 | 33.38 | **35.02** |
| | $\beta$=0.5 | 36.47 | 38.29 | 39.66 | 39.42 | **41.48** |
| | $\beta$=1 | 42.95 | 46.87 | 47.82 | 44.66 | **47.99** |
| | $S$=3 | 77.54 | 80.45 | 82.97 | 81.53 | **83.19** |
| | $S$=5 | 73.6 | 73.05 | 75.44 | 74.78 | **77.13** |
| | $S$=10 | 58.06 | 58.59 | 62.06 | 58.94 | **64.45** |
| **PACS** | $M$=4 | 59.13 | 61.9 | 63.79 | 64.21 | **68.05** |
| | $M$=8 | 58.24 | 61.16 | 62.98 | 63.24 | **65.82** |
| | $M$=20 | 59.23 | 59.98 | 61.56 | 62.49 | **65.05** |

Table 3: Accuracy(%) of the Adience Faces, with (brackets) inside indicating the improvement rate relative to the Local, where the Gender (2) and Age (8) rows show the average test accuracy.

| | Model | Local | FedAvg-c | FedSAK |
|---|---|---|---|---|
| M=2 1 : 1 | Gender (2) | 86.19 | 86.34 (+0.17%) | **88.78** (+3.00%) |
| | Age (8) | 58.21 | 65.09 (+11.82%) | **65.13** (+11.89%) |
| | Total | 72.2 | 75.71 (+4.86%) | **76.95** (+6.58%) |
| M=10 1 : 1 | Gender (2) | 85.69 | 86.02 (+0.39%) | **88.51** (+3.29%) |
| | Age (8) | 58.82 | 63.44 (+7.85%) | **63.54** (+8.02%) |
| | Total | 72.26 | 74.73 (+3.42%) | **76.03** (+5.22%) |
| M=15 1 : 2 | Gender (2) | 85.96 | 82.9 (-3.56%) | **87.01** (+1.22%) |
| | Age (8) | 58.13 | **65.31** (+12.35%) | 65.19 (+12.15%) |
| | Total | 67.41 | 71.17 (+5.58%) | **72.46** (+7.49%) |
| M=15 2 : 1 | Gender (2) | 86.36 | 87.04 (+0.79%) | **88.2** (+2.13%) |
| | Age (8) | 58.23 | 60.9 (+4.59%) | **63.51** (+9.07%) |
| | Total | 76.99 | 78.32 (+1.73%) | **79.97** (+3.87%) |

**Task Heterogeneity** in FL is commonly overlooked but objectively exists in reality, where clients typically train different tasks on a similar dataset. We adopt a large-scale face image dataset, AdienceFaces, which contains gender and age group labels for each person. Specifically, we set up 2, 10, and 15 clients to achieve gender classification and age classification, in which the ratio of heterogeneous tasks are 1:1, 1:2, and 2:1. Under the TH scenario, our shared layers consists of the feature extractor, i.e. $w_i = \varphi_i$. Since there are currently no FL methods that can handle task heterogeneity, we use FedAvg-c to denote clients only uploading the feature extractor and aggregating with FedAvg. Table 3 reports our results. It can be seen that training only with local data is less effective than FL, indicating task heterogeneity is meaningful in the FL setting. We can see that FedSAK achieves the best results under all settings when the task distribution is balanced. When the heterogeneous task distribution is skewed amongst clients, FedAvg-c will be biased towards clients with a larger task proportion. For example, with 15 clients, 5 doing 2-class and 10 doing 8-class, although FedAvg-c performs slightly better on 8-class than FedSAK, its performance on 2-class clients is worse than Local. The confusion matrix in Figure 4 visualizes the results of our classification of the 2 TH clients, and we can see that our method can achieve significant results.

### 6.3 Parameter Experiment

We first evaluated the impact of the trade-off parameters on different heterogeneous setups, Figure 5 depicts the performance of FedSAK on 3 heterogeneous tasks across datasets as $\lambda$ varies. The $\lambda$ controls the extent of coupling among client models, with larger $\lambda$ indicating more emphasis on sharing information among parameters, while smaller $\lambda$ focuses models on utilizing their own data. It can be observed that the optimal $\lambda$ value differs across the heterogeneous tasks. In the data heterogeneity scenario, since models are homogeneous, larger $\lambda$ values yield better performance. In contrast, in the task heterogeneity setting where client differences are greater, smaller $\lambda$ produces improved results. Furthermore, when $\lambda$ is too large, performance decreases in all heterogeneous scenarios. In summary, appropriately tuning $\lambda$ allows balancing between customized local learning and collaborative multi-task training for each heterogeneous scenario.

Additionally, we tested the sensitivity of FedSAK's trade-off parameter $\lambda$, learning rate $\eta$ ($\eta = \eta_\theta = \eta_w$), and local iteration number $E$ on CIFAR-10 ($M$=10, $S$=3). As Figure 6 shows, our model converges under all settings. The convergence curve fluctuates as $\lambda$ changes, but is smoother for learning rate and local iterations, indicating FedSAK is not sensitive to the learning rate and

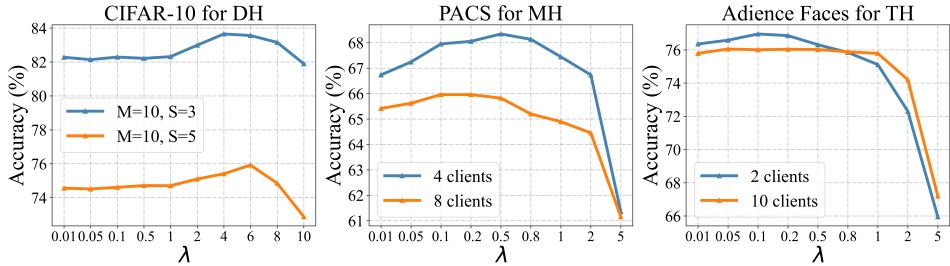

Figure 5: Test accuracy of parameter $\lambda$ in different scenarios.

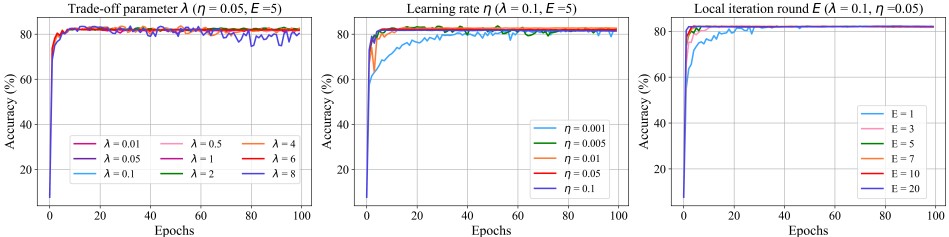

Figure 6: Sensitivity of model parameters.

local iterations. We also find increasing local iterations speeds up convergence, aligning with our theoretical derivations.

### 6.4 Computing and Communication Overhead

We plotted the per-epoch time for each method under the data heterogeneity setting with 100 clients on the MNIST dataset, as shown in Figure 7. The figure shows that personalized methods Ditto and pFedMe spend more time per epoch than most methods, due to the need to train additional personalized models. FedMD, which uses knowledge distillation, also has high time costs because the student and teacher models need to collaborate. Compared to most baselines, FedSAK reduces communication overhead on smaller models by track

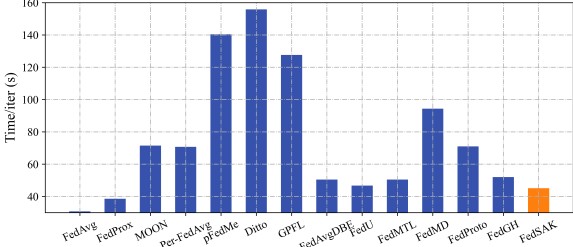

Figure 7: Running time per epoch for each method on the MNIST.

norm regularization updates for local model gradients. It is worth noting that since FedSAK performs tensor track norm, the computational complexity is $O(\min_k d_k^2 \prod_{i \neq k}^{p} d_i)$, which seems to significantly increase with the model's dimensionality, representing a limitation of FedSAK. Nevertheless, FedSAK can balance communication overhead through a flexible upload of shared layer structures. To further demonstrate the communication overhead of FedSAK, we tested it using ResNet18 in the Appendix, as shown in Table 5. It can be seen that FedSAK can also effectively cope with large-scale modeling scenarios.

## 7 Conclusion

In this work, we propose a novel FMTL framework based on tensor trace norm to address challenging federated scenarios with data, model, and task heterogeneity. The method facilitates modeling associations and dependencies among client tasks by flexibly selecting model shared layer structures and uploading them to the server for tensor trace norm regularization. This enables useful knowledge transfer across clients to improve model performance on each task. We conduct comprehensive analyses on the efficacy of the method from both theoretical and experimental perspectives.

## Acknowledgments and Disclosure of Funding

The research is supported by the National Key R&D Program of China (2023YFB2703700), the National Natural Science Foundation of China (62176269), the Guangzhou Science and Technology Program (2023A04J0314), the National Natural Science Foundation of China under Grant (62302537).

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

---

**Algorithm 1:** FedSAK

---

**Input**: total number of clients $M$, number of rounds $T$; learning rate $\eta_\theta, \eta_w$, hyper-parameter for loss $\lambda$.

Randomly initialize the heterogeneous local models $\left[\theta_1^0, \ldots, \theta_M^0\right]$ and share layer $\widetilde{w}^0$.

**for** $t = 1$ **to** $T$ **do**

    **// Clients Side**:

    **for** *each batch in* $\mathcal{B}_i$ **do**

        1: Receive the share layer $\widetilde{w}_i^{t-1}$ broadcast by the server;

        2: Update the local model: $\widetilde{\theta}_i^t$ ;

        3: Perform local training: $\theta_i^t \leftarrow \widetilde{\theta}_i^t - \eta_\theta \nabla \mathcal{L}_c\left(\widetilde{\theta}_i^t; \mathcal{B}_i\right)$;

    **end**

    4: Upload the global shared layers $w_i^t$ to the server.

    **// Server Side**:

    1: Receive the global shared layers $w_i^t$ from $M$ clients and stack the parameters into $L$ tensors via Eq.(4);

    2: Calculated loss :$\mathcal{L}_r = \lambda \sum_{l=1}^{L} \sum_{k=1}^{p} \left\| \mathcal{W}_{(k)}^{l,t} \right\|_*$;

    3: Calculate gradient to update client parameters:

    **for** $i \in M$ **do**

        $\widetilde{w}_i^t \leftarrow w_i^t - \eta_w \nabla \mathcal{L}_r$;

    **end**

    4: Broadcast the updated global shared layers $w_i$ to the $i$-th client.

**end**

**Return** Personalized heterogeneous private models for all clients: $\left[\theta_1^T, \theta_2^T, \ldots, \theta_M^T\right]$.

---

# A  Supplementary Experiments

## A.1  Training Setup

See Algorithm 1 for details of our algorithm.

For local optimization, all methods utilize mini-batch SGD, with the number of local epochs set to $E$=5 and the batch size selection range is $\{16, 20, 32\}$ per client. The number of global communication rounds is uniformly set to $t$=100 across all datasets, which is sufficient for the convergence of the FL methods. We report the average test accuracy across all clients as the evaluation metric after convergence. We also tune special hyperparameters for baselines and report the optimal results. We ran these experiments using 4 NVIDIA GeForce RTX 4090 GPUs for all methods.

## A.2  Model Setup

**Data heterogeneity.** In the data heterogeneity scenario, to explore only the impact of data heterogeneity, we set up each client to have the same model structure. On the HumA and MNIST datasets, we used 2 fully connected layers, i.e., $input \times 100 \times classnumber$. On the CIFAR dataset, we used a CNN model with 2 convolutional layers and 2 fully connected layers. Where the convolutional layer kernel is $5 * 5$ and the filter distribution is 32 or 64, i.e. $(5 \times 5, 32)$ and $(5 \times 5, 64)$. Connecting 2 fully connected layers, i.e. $input \times 512 \times class\ number$. While communicating with the server, these model structures are uploaded to the server for tensor trace norm constraints.

**Model heterogeneity.** In the model heterogeneity setup, we vary the number of filters in the convolutional layer and the dimensionality of the fully connected layer in the CNN model to obtain four heterogeneous models The detailed design of each model is shown in Table 4. In the model heterogeneity experiments, we select different heterogeneous models for each client in turn. In this case, we only uploaded the prediction head to the server communication.

**Task heterogeneity.** We adopt the same network structure as the CIFAR dataset in the data heterogeneity scenario. And consider the last fully connected layer as the prediction head, i.e., $512 \times class\ number$. Therefore, in the task heterogeneity scenario, we uploaded a feature extractor

Table 4: The structure of the four heterogeneous CNN models in model heterogeneity, with the client model selected in order by id ($p$ for padding, $s$ for stride).

| | layer name | Model-1 | Model-2 | Model-3 | Model-4 |
|---|---|---|---|---|---|
| **Feature extractor** | Conv1 | 16, 5×5 $p$=2, $s$=1 | 32, 3×3 $p$=1, $s$=2 | 32, 5×5 $p$=0, $s$=1 | 32, 3×3 $p$=0, $s$=1 |
| | Pool | (2, 2) | (2, 2) | (2, 2) | (2, 2) |
| | Conv2 | 32, 3×3 $p$=1, $s$=1 | 64, 2×2 $p$=1, $s$=2 | 64, 5×5 $p$=0, $s$=1 | 64, 5×5 $p$=0, $s$=1 |
| | Pool | (2, 2) | - | (2, 2) | (2, 2) |
| | Conv3 | 64, 2×2 $p$=1, $s$=2 | - | - | - |
| | FC1 | 512 | 512 | 512 | 512 |
| **Prediction head** | FC2 | 10 / 7 | 10 / 7 | 10 / 7 | 10 / 7 |

consisting of 2 convolutional layers i.e. $(5 \times 5, 32)$ and $(5 \times 5, 64)$ and 1 fully connected layer i.e. $input \times 512$ to interact with other clients.

### A.3  Ablation Experiment

To emphasize the effectiveness of FedSAK's knowledge transfer using the tensor trace norm, we conducted ablation experiments. Due to model-specific design choices, when the model does not communicate, the model degrades to the Local state. When the shared layer uploads are weighted and aggregated only on the server side, our approach will degrade to the standard FedAvg algorithm. It is worth noting that FedAvg cannot do weighted averaging on the whole model in model heterogeneous and task heterogeneous scenarios. Therefore we only performed FedAvg computation on the uploaded shared layer. The experimental results are shown in Figure 8, where it can be seen that FedSAK for client knowledge transfer learning using tensor trace paradigms yields better results.

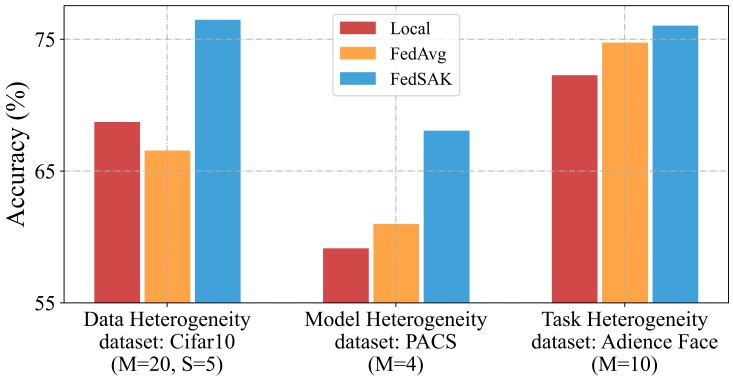

Figure 8: Sensitivity of model parameters.

### A.4  Computing Overhead

In Section 6.4, we report a FedSAK computational complexity of $O(\min_k d_k^2 \prod_{i \neq k}^p d_i)$ and compare the communication overhead of FedSAK with the baseline method on the MNIST dataset. The computational complexity of our method does increase with network size. Due to its flexibility in uploading shared structures, FedSAK can easily handle larger networks compared to methods that upload the entire model (e.g., FedAvg). To further emphasize the effectiveness of FedSAK on complex networks, we tested it using ResNet18 on the CIFAR-10 dataset. We set up a total of 20 clients, each containing 5 labels. In the FedSAK model, we shared the fully connected layer behind.

Table 5: The accuracy, time consumption, and memory usage of each method running ResNet18 on the CIFAR-10 dataset.

| Model | Accuracy (%) | Total Time Consumption | Used Memory |
|---|---|---|---|
| FedAvg | 68.05 | 6628s | 1.75 G |
| FedProx | 59.55 | 6729s | 2.58 G |
| MOON | 68.52 | 9866s | 2.63 G |
| Per-FedAvg | 73.35 | 10822s | 1.75 G |
| pFedMe | 75.84 | 57406s | 2.63 G |
| Ditto | 77.98 | 19366s | 3.42 G |
| GPFL | 78.59 | 18749s | 1.83 G |
| FedAvgDBE | 75.54 | 7898s | 1.75 G |
| FedMTL | 73.68 | 11757s | 2.58 G |
| FedU | 75.98 | 9584s | 2.58 G |
| FedMD | 73.74 | 16084s | 3.50 G |
| FedProto | 78.34 | 12076s | 1.75 G |
| FedGH | 75.95 | 7508s | 1.71 G |
| FedSAK | 77.69 | 7118s | 0.918 G |

Table 5 reports the accuracy, time consumption, and memory usage of each method running ResNet18 on the CIFAR10 dataset. It can be seen that when only fully connected layer is uploaded, our model is still advantageous compared to most methods. In addition, its time consumption and memory occupation are much smaller than other methods, which greatly demonstrates the flexibility of our method.

## B Datasets

### B.1 Dataset Description

- **Human Activity [41]:** Mobile phone accelerometer and gyroscope data collected from 30 individuals, performing one of six activities: walking, walking-upstairs, walking-downstairs, sitting, standing, lying-down. We use the provided 561-length feature vectors of time and frequency domain variables generated for each instance. We model each individual as a separate task and predict between sitting and the other activities.

- **MNIST [42]:** A handwritten digit dataset includes 10 labels and 70,000 instances. The whole dataset is distributed to N = 100 clients. Each client has a different local data size and consists of 2 labels.

- **CIFAR-10 & CIFAR-100 [43]:** CIFAR-10 consists of 60000, $32 \times 32$ color images in 10 classes, with 6,000 images per class. Similar to CIFAR-10, CIFAR-100 has 100 classes, with 600 images per class. We partition the dataset into $M$ clients, with each client assigned $S$ labels. Datasets are split randomly with 75% and 25% for training and testing, respectively.

- **PACS [44]:** PACS is a challenging heterogeneous dataset with large domain discrepancies. The PACS dataset has a total of 9,991 images, each of size $3 \times 227 \times 227$. The dataset consists of 4 distinct domains: art painting, cartoon, photo, and sketch. Each domain contains 7 classes: Dog, Elephant, Giraffe, Guitar, Horse, House, Person. The picture of this dataset is shown in Figure 9.

- **Adience Faces [45]:** Adience Faces is a large-scale face image dataset with labels for gender and age group of each individual. This enables the dataset to be used for two tasks: (i) gender classification into two classes (male and female), and (ii) age group classification into eight groups (0-2, 4-6, 8-12, 15-20, 25-32, 38-48, 48-53, 60-100 years old). We use this dataset to evaluate heterogeneous task federated learning, where each client is assigned one of the tasks for training.

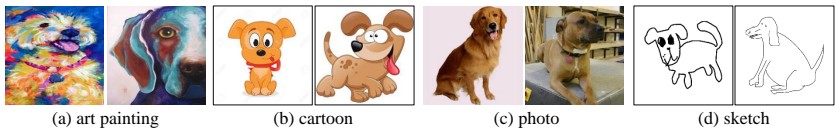

(a) art painting  (b) cartoon  (c) photo  (d) sketch

Figure 9: The PACS datasets.

## B.2   Data Distribution Visualization

We visualized the data in order to clearly show the client's training data. The red circle represents the percentage of data, the larger the circle, the more data the client has of that type.

In the experiments with data heterogeneity, we follow the setup of the FMTL reference [7, 8], where each client contains a different number of partially labeled classes. Thus in our data heterogeneity setup, the **data distribution** of clients on Human Activity, MNIST, CIFAR-10, and CIFAR-100 are shown in Figure 10 - 13 .

The CIFAR-10 and PACS datasets were the main ones we used in the experiments on **model heterogeneity**. In CIFAR-10 we employed the Dillikerley distribution to divide the heterogeneous data see Figure 14 and the labeled division data see Figure 12, respectively. PACS contains data from 4 different domains, which is more suitable for model heterogeneity in real scenarios. Since this data itself has a high degree of heterogeneity, we did not have a heterogeneous division of the client data label distribution, the data distribution is shown in Figure 15.

In the experiments of **task heterogeneity**, we use Adience Facs dataset and each client samples similar data for different training tasks, and its data distribution is shown in Figure 16.

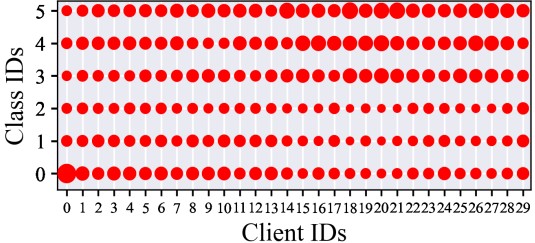

Figure 10: The data distribution on Human Activity.

Figure 11: The data distribution on MNIST.

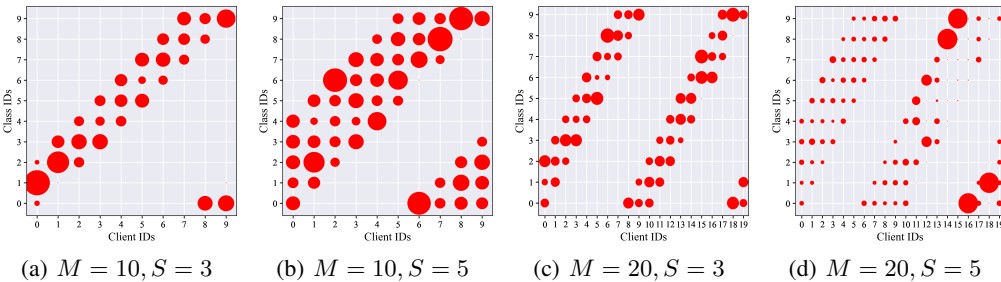

(a) $M = 10, S = 3$     (b) $M = 10, S = 5$     (c) $M = 20, S = 3$     (d) $M = 20, S = 5$

Figure 12: The data distribution of all clients on CIFAR-10.

## C   Convergence Analysis

To analyze the convergence of FedSAK, we follow [15, 16] and make the following assumptions C.1: Define $t$ as the current communication round, $e \in \{0, 1, \cdots, E\}$ as the number of local iterations, where $E$ denotes the maximum number of local iterations. Thus, $(tE + e)$ represents the e-th iteration

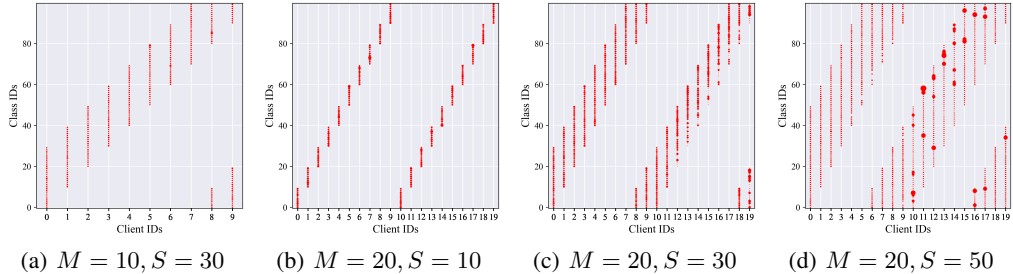

(a) $M = 10, S = 30$    (b) $M = 20, S = 10$    (c) $M = 20, S = 30$    (d) $M = 20, S = 50$

Figure 13: The data distribution of all clients on CIFAR-100.

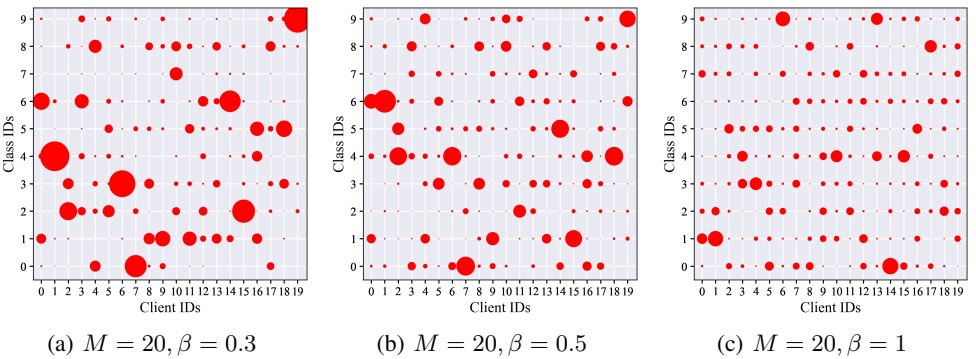

(a) $M = 20, \beta = 0.3$    (b) $M = 20, \beta = 0.5$    (c) $M = 20, \beta = 1$

Figure 14: The data distribution of all clients on CIFAR-10.

in the $(t+1)$-th communication round. The $(t+0)$ denotes that at the beginning of the $(t+1)$-th round, the client uses the global shared layers gradients from round $t$ to update the local shared layer parameters. Note that $(tE + E)$ corresponds to the last iteration in round $(t+1)$.

## C.1 Assumption

**Assumption 1 (Lipschitz Smoothness)** *The $i$-th client's local model loss function $\mathcal{L}$ is Lipschitz continuous with Lipschitz constant $K$, and $0 \leq \mathcal{L} \leq a$ with $\mathcal{L}(0) = 0$, i.e.,*

$$\|\nabla\mathcal{L}_{t_1} - \nabla\mathcal{L}_{t_2}\|_2 \leq K\|\theta_i^{t_1} - \theta_i^{t_2}\|_2, \\ \forall t_1, t_2 > 0, i \in \{1, 2, \ldots, M\}. \tag{16}$$

**Assumption 2 (Unbiased Gradient and Bounded Variance)** *The random gradient $g_i^t = \nabla\mathcal{L}_t\left(\theta_i^t; \mathcal{B}_i^t\right)$ of each client's local model is unbiased, where $\mathcal{B}$ is a batch of local data, i.e.,*

$$\mathbb{E}_{\mathcal{B}_i^t \subseteq n_i}\left[g_i^t\right] = \nabla\mathcal{L}(\theta_i^t) = \nabla\mathcal{L}_t, \forall i \in \{1, 2, \ldots, M\}, \tag{17}$$

*and the variance of random gradient $g_k^t$ is bounded by:*

$$\mathbb{E}_{\mathcal{B}_i^t \subseteq n_i}\left[\left\|\nabla\mathcal{L}_t\left(\theta_i^t; \mathcal{B}_i^t\right) - \nabla\mathcal{L}_t\left(\theta_i^t\right)\right\|_2^2\right] \leqslant \sigma^2. \tag{18}$$

**Assumption 3 (Bounded Variance of the Shared Layers)** *The variance of the shared layers $w_i$ of the model $\theta_i$ trained by client $i$ with local data $n_i$, and the server tensor low-rank constrained update to the shared layers $\widetilde{w}_i$ are bounded, i.e.,*

*parameter bounded:* $\mathbb{E}\left[\|w_i - \widetilde{w}_i\|_2^2\right] \leqslant \varepsilon^2$,

*gradient bounded:* $\mathbb{E}\left[\|\nabla\mathcal{L}_r\left(w_i\right) - \nabla\mathcal{L}_r(\widetilde{w}_i)\|_2^2\right] \leqslant \omega^2$.

Based on the above assumptions, due to our same local training, Tan *et al.* [15] and Yi *et al.* [16] deduce that Lemma 1 and 2 still holds. For notational simplicity, we set $\eta = \eta_\theta = \eta_w$.

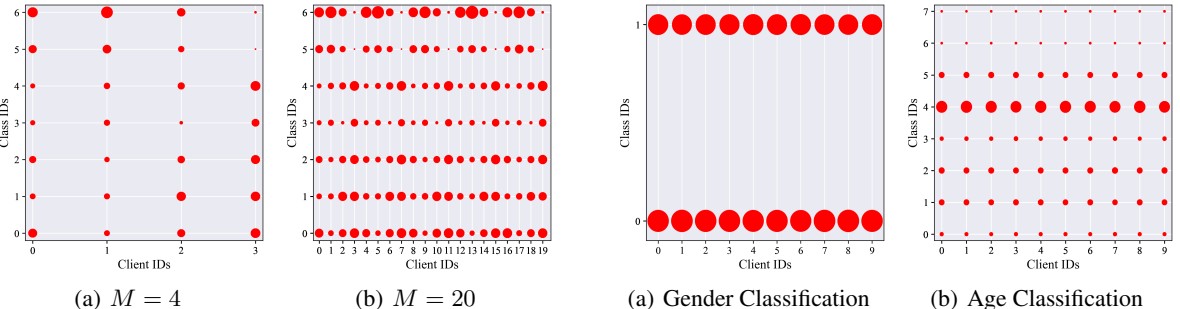

(a) $M = 4$      (b) $M = 20$      (a) Gender Classification      (b) Age Classification

Figure 15: The data distribution of all clients on PACS in practical settings with 4 and 20 clients, respectively.

Figure 16: The data distribution of all clients on Adience Faces.

## C.2    Proof for Lemma 1

**Proof.** For arbitrary clients, we have $\theta^{t+1} = \theta^t - \eta g^t$, then

$$
\begin{aligned}
\mathcal{L}_{tE+1} &\le \mathcal{L}_{tE+0} + \langle \nabla \mathcal{L}_{tE+0}, (\theta^{tE+1} - \theta^{tE+0}) \rangle \\
&\quad + \frac{K}{2} \| \theta^{tE+1} - \theta^{tE+0} \|_2^2 \\
&= \mathcal{L}_{tE+0} - \eta \langle \nabla \mathcal{L}_{tE+0}, g^{tE+0} \rangle + \frac{K}{2} \| \eta g^{tE+0} \|_2^2,
\end{aligned}
\tag{19}
$$

Taking expectation of both sides of the above equation on the random variable $\mathcal{B}$, we have

$$
\begin{aligned}
\mathbb{E}[\mathcal{L}_{tE+1}] &\le \mathcal{L}_{tE+0} - \eta \mathbb{E}[\langle \nabla \mathcal{L}_{tE+0}, g^{tE+0} \rangle] + \frac{K\eta^2}{2} \mathbb{E}[\| g^{tE+0} \|_2^2] \\
&= \mathcal{L}_{tE+0} - \eta \| \nabla \mathcal{L}_{tE+0} \|_2^2 + \frac{K\eta^2}{2} \mathbb{E}[\| g_{i,tE+0} \|_2^2] \\
&\le \mathcal{L}_{tE+0} - \eta \| \nabla \mathcal{L}_{tE+0} \|_2^2 + \frac{K\eta^2}{2} (\| \nabla \mathcal{L}_{tE+0} \|_2^2 + \mathrm{Var}(g^{i,tE+0})) \\
&= \mathcal{L}_{tE+0} - (\eta - \frac{K\eta^2}{2}) \| \nabla \mathcal{L}_{tE+0} \|_2^2 + \frac{K\eta^2}{2} \mathrm{Var}(g^{i,tE+0}) \\
&\le \mathcal{L}_{tE+0} - (\eta - \frac{K\eta^2}{2}) \| \nabla \mathcal{L}_{tE+0} \|_2^2 + \frac{K\eta^2}{2} \sigma^2,
\end{aligned}
\tag{20}
$$

where $\mathrm{Var}(x) = \mathbb{E}[x^2] - (\mathbb{E}[x])^2$. Take expectation of $\theta$ on both sides. Then, by telescoping of $E$ steps, we have,

$$
\mathbb{E}[\mathcal{L}_{(t+1)E}] \le \mathcal{L}_{tE+0} - (\eta - \frac{K\eta^2}{2}) \sum_{e=0}^{E} \| \nabla \mathcal{L}_{tE+e} \|_2^2 + \frac{KE\eta^2}{2} \sigma^2.
\tag{21}
$$

## C.3    Proof for Lemma 2

Until the proof, we denote $q$ to indicate that the local model has not been used as a parameter for uploading the server side as a shared layer, i.e., $q = \theta - w$. Note that the client symbol $i$ is omitted since it is used for arbitrary clients.

**Proof.**

$$\mathcal{L}_{(t+1)E+0} = \mathcal{L}_{(t+1)E} + \mathcal{L}_{(t+1)E+0} - \mathcal{L}_{(t+1)E}$$
$$=\mathcal{L}_{(t+1)E} + \mathcal{L}\left(\left(q^{t+1}, \widetilde{w}^{t+1}\right); \boldsymbol{x}, y\right) - \mathcal{L}\left(\left(q^{t+1}, w^{t+1}\right); \boldsymbol{x}, y\right)$$
$$\leqslant \mathcal{L}_{(t+1)E} + \left\langle \nabla\mathcal{L}\left(\left(q^{t+1}, w^{t+1}\right)\right), \left(\left(q^{t+1}, \widetilde{w}^{t+1}\right) - \left(q^{t+1}, w^{t+1}\right)\right)\right\rangle$$
$$+ \frac{K\lambda}{2}\left\|\left(q^{t+1}, \widetilde{w}^{t+1}\right) - \left(q^{t+1}, w^{t+1}\right)\right\|_2^2$$
$$\leqslant \mathcal{L}_{(t+1)E} + \frac{K\lambda}{2}\left\|\left(q^{t+1}, \widetilde{w}^{t+1}\right) - \left(q^{t+1}, w^{t+1}\right)\right\|_2^2$$
$$\leqslant \mathcal{L}_{(t+1)E} + \frac{K\lambda}{2}\left\|\widetilde{w}^{t+1} - w^{t+1}\right\|_2^2 \tag{22}$$
$$=\mathcal{L}_{(t+1)E} + \frac{K\lambda}{2}\left\|\widetilde{w}^t - \eta\nabla\mathcal{L}\left(\widetilde{w}^t\right) - w^t + \eta\nabla\mathcal{L}\left(w^t\right)\right\|_2^2$$
$$= \mathcal{L}_{(t+1)E} + \frac{K\lambda}{2}\left\|\widetilde{w}^t - w^t + \eta\left(\nabla\mathcal{L}\left(w^t\right) - \nabla\mathcal{L}\left(\widetilde{w}^t\right)\right)\right\|_2^2$$
$$\leqslant \mathcal{L}_{(t+1)E} + \frac{K\lambda}{2}\left\|\eta\left(\nabla\mathcal{L}\left(w^t\right) - \nabla\mathcal{L}\left(\widetilde{w}^t\right)\right)\right\|_2^2$$
$$= \mathcal{L}_{(t+1)E} + \frac{\eta K\lambda}{2}\left\|\left(\nabla\mathcal{L}\left(w^t\right) - \nabla\mathcal{L}\left(\widetilde{w}^t\right)\right)\right\|_2^2.$$

The lemma is proved.

Taking expectation of both sides of the above equation on the random variable $\mathcal{B}$, we have

$$\mathbb{E}\left[\mathcal{L}_{(t+1)E+0}\right] \leqslant \mathbb{E}\left[\mathcal{L}_{(t+1)E}\right] + \frac{\eta K\lambda}{2}\mathbb{E}\left[\left\|\left(\nabla\mathcal{L}\left(w^t\right) - \nabla\mathcal{L}\left(\widetilde{w}^t\right)\right)\right\|_2^2\right]$$
$$\leqslant \mathbb{E}\left[\mathcal{L}_{(t+1)E}\right] + \frac{\eta K\lambda\omega^2}{2}. \tag{23}$$

### C.4 Proof for Theorem 1

Before presenting the proof for Theorem 1, then prove the following theorem.

**Theorem 4** *Based on the above assumptions, the expectation of the loss of an arbitrary client's local model before the start of a round of local iteration satisfies*

$$\mathbb{E}\left[\mathcal{L}_{(t+1)E+0}\right] \leqslant \mathcal{L}_{tE+0} - \left(\eta - \frac{K\eta^2}{2}\right)\sum_{e=0}^{E}\|\mathcal{L}_{tE+e}\|_2^2$$
$$+ \frac{\eta K\left(E\eta\sigma^2 + \lambda\omega^2\right)}{2}. \tag{24}$$

**Proof.** Substituting Lemma 1 into the second term on the right-hand side of Lemma 2 proves it.

Then we can prove Theorem 1 as follows:
**Proof.** Transform the form of Theorem 4 into

$$\sum_{e=0}^{E}\|\mathcal{L}_{tE+e}\|_2^2 \leqslant \frac{\mathcal{L}_{tE+0} - \mathbb{E}\left[\mathcal{L}_{(t+1)E+0}\right] + \frac{\eta K\left(E\eta\sigma^2 + \lambda\omega^2\right)}{2}}{\eta - \frac{K\eta^2}{2}}. \tag{25}$$

Take expectations of model $\theta$ on both sides, we have:

$$\sum_{e=0}^{E}\mathbb{E}\left[\|\mathcal{L}_{tE+e}\|_2^2\right] \leqslant \frac{\mathbb{E}\left[\mathcal{L}_{tE+0}\right] - \mathbb{E}\left[\mathcal{L}_{(t+1)E+0}\right] + \frac{\eta K\left(E\eta\sigma^2 + \lambda\omega^2\right)}{2}}{\eta - \frac{K\eta^2}{2}}. \tag{26}$$

Since $\sum_{t=1}^{T} \left( \mathbb{E}\left[\mathcal{L}_{tE+0}\right] - \mathbb{E}\left[\mathcal{L}_{(t+1)E+0}\right] \right) \leqslant \mathcal{L}_{t=1} - \mathcal{L}^*$, for each round:

$$\frac{1}{TE} \sum_{t=0}^{T-1} \sum_{e=0}^{E} \mathbb{E}\left[\left\|\mathcal{L}_{tE+e}\right\|_2^2\right]$$

$$\leqslant \frac{\frac{1}{TE}\sum_{t=0}^{T-1}\left(\mathbb{E}\left[\mathcal{L}_{tE+0}\right] - \mathbb{E}\left[\mathcal{L}_{(t+1)E+0}\right]\right) + \frac{\eta K\left(E\eta\sigma^2+\lambda\omega^2\right)}{2}}{\eta - \frac{K\eta^2}{2}}$$

$$\leqslant \frac{\frac{1}{TE}\left(\mathcal{L}_{t=1} - \mathcal{L}^*\right) + \frac{\eta K\left(E\eta\sigma^2+\lambda\omega^2\right)}{2}}{\eta - \frac{K\eta^2}{2}} \qquad (27)$$

$$= \frac{2\left(\mathcal{L}_{t=1} - \mathcal{L}^*\right) + \eta K T E\left(E\eta\sigma^2 + \lambda\omega^2\right)}{TE\left(2\eta - K\eta^2\right)}$$

$$= \frac{2\left(\mathcal{L}_{t=1} - \mathcal{L}^*\right)}{TE\eta\left(2 - K\eta\right)} + \frac{K\left(E\eta\sigma^2 + \lambda\omega^2\right)}{E(2 - K\eta)}.$$

Given any $\epsilon > 0$ the above equation satisfies

$$\frac{2\left(\mathcal{L}_{t=1} - \mathcal{L}^*\right)}{TE\eta\left(2 - K\eta\right)} + \frac{K\left(E\eta\sigma^2 + \lambda\omega^2\right)}{E(2 - K\eta)} \leqslant \epsilon. \qquad (28)$$

Then, we can obtain:

$$T \geqslant \frac{2\left(\mathcal{L}_{t=1} - \mathcal{L}^*\right)}{E\eta\epsilon\left(2 - K\eta\right) - \eta K\left(E\eta\sigma^2 + \lambda\omega^2\right)}. \qquad (29)$$

Since $T > 0, \mathcal{L}_{t=1} - \mathcal{L}^* > 0$, we can further derive:

$$E\eta\epsilon\left(2 - K\eta\right) - \eta K\left(E\eta\sigma^2 + \lambda\omega^2\right) > 0, \qquad (30)$$

i.e.,

$$\eta < \frac{2E\epsilon - K\lambda\omega^2}{KE\left(\epsilon + \sigma^2\right)}. \qquad (31)$$

## D  Excess risk bound

### D.1  Proof for Lemma 3

**Proof.** By following [40], we have

$$\|\mathcal{W}\|_{*\star} = \inf_{\sum_{k\neq\varnothing, k\subset[p]}\mathcal{W}^{(k)}=\mathcal{W}} \max_k \left\|\mathcal{W}_{(k)}^{(k)}\right\|_\infty, \qquad (32)$$

where $[p]$ denotes a set of positive integers no larger than $p$. Since we can take any $\mathcal{W}^{(k)}$ to equal $\mathcal{W}$, the norm can be upper bounded as follows:

$$\|\mathcal{W}\|_{*\star} \leq \min_k \left\|\mathcal{W}_{(k)}\right\|_\infty. \qquad (33)$$

### D.2  Proof for Theorem 2

**Proof.** Based on Lemma 3, given that the minimum expectation across $k$ can be confined to an upper limit by the minimum of the expected values, it follows that

$$\mathbb{E}\|\mathcal{W}\|_{*\star} \leq \mathbb{E}\min_k \left\|\mathcal{W}_{(k)}\right\|_\infty \leq \min_k \mathbb{E}\left\|\mathcal{W}_{(k)}\right\|_\infty. \qquad (34)$$

Referring to Theorem 6.1 in Reference [47], we upper bound for each expectation that

$$\Pr\{\left\|\mathcal{W}_{(k)}\right\|_\infty \geq t\} \leq \begin{cases} D_k exp(-3t^2/8\sigma_k^2), & \text{for } t \leq \sigma_k^2/R_k, \\ D_k exp(-3t/8R_k), & \text{for } t \geq \sigma_k^2/R_k, \end{cases} \qquad (35)$$

and

$$\mathbb{E}\left\|\mathcal{W}_{(k)}\right\|_\infty \leq C(\sigma_k\sqrt{lnD_k} + R_k lnD_k), \qquad (36)$$

where $C$ is an absolute constant, and $\mathcal{Z}^{i,j}$ is a $d_1 \times \cdots \times d_{p-1} \times d_p$ zero tensor with only the ith slice along the last axis equal to $\frac{1}{n}\sigma_i^j \mathbf{x}_i^j$, and in addition, $R_k$ satisfies $R_k \geq ||\mathcal{Z}_{(k)}^{i,j}||_\infty$, therefore:

$$\sigma_k^2 = \max\left(\left\|\sum_{i=1}^M \sum_{j=1}^n \mathbb{E}[\mathcal{Z}_{(k)}^{i,j}(\mathcal{Z}_{(k)}^{i,j})^T]\right\|_\infty , \left\|\sum_{i=1}^M \sum_{j=1}^n \mathbb{E}[(\mathcal{Z}_{(k)}^{i,j})^T \mathcal{Z}_{(k)}^{i,j}]\right\|_\infty\right). \tag{37}$$

Since the Frobenius norm of a matrix is larger than its spectral norm, $||\mathcal{Z}_{(k)}^{i,j}||_\infty \leq \frac{1}{n}$ and we simply set $R_k = \frac{1}{n}$. For $\sigma_k$, we obtain

$$\mathbb{E}[\sum_{j=1}^N \mathcal{Z}_{(k)}^{i,j}(\mathcal{Z}_{(k)}^{i,j})^T] = \frac{1}{n}\mathbf{C}_{k-\{p\}} \preceq \frac{\kappa}{nd}\mathbf{I}, \tag{38}$$

where $\kappa > 0$ is given constant. This means:

$$\left\|\sum_{i=1}^M \sum_{j=1}^n \mathbb{E}[\mathcal{Z}_{(k)}^{i,j}(\mathcal{Z}_{(k)}^{i,j})^T]\right\|_\infty \leq \frac{\kappa M}{nd}. \tag{39}$$

Similarly, we have

$$\mathbb{E}[\sum_{j=1}^n (\mathcal{Z}_{(k)}^{i,j})^T \mathcal{Z}_{(k)}^{i,j}] = \mathrm{diag}\left(\frac{\mathrm{tr}\left(\mathbf{C}_{k-\{p\}}\right)}{n}\right) \preceq \frac{\kappa}{nd}\mathbf{I}, \tag{40}$$

where $\mathrm{tr}(\cdot)$ denotes the trace on a matrix and $\mathrm{diag}(\cdot)$ converts a vector or scalar to a diagonal matrix. This inequality implies

$$\left\|\sum_{i=1}^M \sum_{j=1}^n \mathbb{E}[(\mathcal{Z}_{(k)}^{i,j})^T \mathcal{Z}_{(k)}^{i,j}]\right\|_\infty \leq \frac{\kappa M}{nd}. \tag{41}$$

Substituting inequalities Eq. (39) and Eq. (41) into Eq. (36), we obtain inequality Eq.(14).

### D.3    Proof for Theorem 3

**Proof.** Note that

$$\mathcal{R}(\hat{\mathcal{W}}) - \mathcal{R}(\mathcal{W}) = \underbrace{\mathcal{R}(\hat{\mathcal{W}}) - \hat{\mathcal{R}}(\hat{\mathcal{W}})}_{r1} + \underbrace{\hat{\mathcal{R}}(\hat{\mathcal{W}}) - \hat{\mathcal{R}}(\mathcal{W})}_{r2} + \underbrace{\hat{\mathcal{R}}(\mathcal{W}) - \mathcal{R}(\mathcal{W})}_{r3} \tag{42}$$

We first establish the upper bound for part $r3$ in Eq. (42). Under Assumption 1, it follows from Hoeffding's inequality [48] that

$$\hat{\mathcal{R}}(\mathcal{W}) - \mathcal{R}(\mathcal{W}) \leq a\sqrt{\frac{log(2/\delta)}{2\sum_i n_i}}, \tag{43}$$

with probability at least $1 - \frac{\delta}{2}$. Since $\hat{\mathcal{W}}$ is the optimal solution of Eq. (12), for the part $r2$, we have

$$\hat{\mathcal{R}}(\hat{\mathcal{W}}) - \hat{\mathcal{R}}(\mathcal{W}) \leq 0. \tag{44}$$

Since $\mathcal{L}$ is bounded such that the perturbation of Eq. (12) to $\mathbf{x}_i^j$ is less than $\frac{a}{Mn_i}$. By McDiarmid's inequality [49], the part $r1$ we have:

$$\mathcal{R}(\hat{\mathcal{W}}) - \hat{\mathcal{R}}(\hat{\mathcal{W}}) \leq \sup_{\|\mathcal{W}\|_* \leq \gamma} \{\mathcal{R}(\mathcal{W}) - \hat{\mathcal{R}}(\mathcal{W})\}$$

$$\leq \mathbb{E}\left[\sup_{\|\mathcal{W}\|_* \leq \gamma} \{\mathcal{R}(\mathcal{W}) - \hat{\mathcal{R}}(\mathcal{W})\}\right] + a\sqrt{\frac{log(2/\delta)}{2Mn}}, \tag{45}$$

with probability at least $1 - \frac{\delta}{2}$. Therefore, plugging Eq. (43-45), into Eq. (42), we deduce

$$
\begin{aligned}
\mathcal{R}(\hat{\mathcal{W}}) - \mathcal{R}(\mathcal{W}) &\leq \mathbb{E}\left[\sup_{\|\mathcal{W}\|_* \leq \gamma} \{\mathcal{R}(\mathcal{W}) - \hat{\mathcal{R}}(\mathcal{W})\}\right] \\
&\quad + a\sqrt{\frac{\log(2/\delta)}{2Mn}} + a\sqrt{\frac{log(2/\delta)}{2\sum_i n_i}} \\
&\leq \mathbb{E}\left[\sup_{\|\mathcal{W}\|_* \leq \gamma} \{\mathcal{R}(\mathcal{W}) - \hat{\mathcal{R}}(\mathcal{W})\}\right] + a\sqrt{\frac{2\log(2/\delta)}{Mn}},
\end{aligned}
\tag{46}
$$

where the second inequality follows from $n_i \geq n$. Now we estimate an upper bound of the expectation in Eq. (46) as:

$$
\begin{aligned}
&\mathbb{E}\left[\sup_{\|\mathcal{W}\|_* \leq \gamma} \{\mathcal{R}(\mathcal{W}) - \hat{\mathcal{R}}(\mathcal{W})\}\right] \\
&\leq \mathbb{E}\sup_{\|\mathcal{W}\|_* \leq \gamma} \left| \frac{1}{M}\sum_{n_i} \mathbb{E}_{(\mathbf{x},y)\sim\mathcal{P}_i}\mathcal{L}(\mathcal{F}_i(\mathcal{W};\mathbf{x}),y) \right. \\
&\qquad\qquad\quad \left. - \frac{1}{M}\sum_{i=1}^{M}\frac{1}{n_i}\sum_{j=1}^{n_i}\mathcal{L}(\mathcal{F}_i(\mathcal{W};\mathbf{x}_i^j),y_i^j) \right| \\
&\leq \mathbb{E}\sup_{\|\mathcal{W}\|_* \leq \gamma} \left| \frac{2}{M}\sum_{i=1}^{M}\frac{1}{n_i}\sum_{j=1}^{n_i}\sigma_i^j\mathcal{L}(\mathcal{F}_i(\mathcal{W};\mathbf{x}_i^j),y_i^j) \right|,
\end{aligned}
\tag{47}
$$

where the expectation is taken over the Rademacher random variables and the training samples. Then, we obtain that

$$
\begin{aligned}
&\mathbb{E}\sup_{\|\mathcal{W}\|_* \leq \gamma} \left| \frac{2}{M}\sum_{i=1}^{M}\frac{1}{n_i}\sum_{j=1}^{n_i}\sigma_i^j\mathcal{L}(\mathcal{F}_i(\mathcal{W};\mathbf{x}_i^j),y_i^j) \right| \\
&\leq \mathbb{E}\sup_{\|\mathcal{W}\|_* \leq \gamma} \left| \frac{2}{M}\sum_{i=1}^{M}\frac{1}{n_i}\sum_{j=1}^{n_i}\sigma_i^j\left[K|y_i^j| + \mathcal{L}(\mathcal{F}_i(\mathcal{W};\mathbf{x}_i^j))\right] \right| \\
&\leq K\mathbb{E}\sup_{\|\mathcal{W}\|_* \leq \gamma} \left| \frac{4}{M}\sum_{i=1}^{M}\frac{1}{n_i}\sum_{j=1}^{n_i}\sigma_i^j\mathcal{F}_i(\mathcal{W};\mathbf{x}_i^j) \right| \\
&\quad + K\mathbb{E}\left| \frac{2}{M}\sum_{i=1}^{M}\frac{1}{n_i}\sum_{j=1}^{n_i}\sigma_i^j|y_i^j| \right| \\
&= \frac{4K}{M}\mathbb{E}\left[\sup_{\|\mathcal{W}\|_* \leq \gamma}|\langle\mathcal{W}_\star,\mathcal{W}\rangle|\right] + K\mathbb{E}\left| \frac{2}{M}\sum_{i=1}^{M}\frac{1}{n_i}\sum_{j=1}^{n_i}\sigma_i^j|y_i^j| \right| \\
&\leq \frac{4K}{M}\mathbb{E}\left[\sup_{\|\mathcal{W}\|_* \leq \gamma}\|\mathcal{W}\|_{**},\|\mathcal{W}\|_*\right] + \frac{2K}{Mn}\mathbb{E}\left| \sum_{i=1}^{M}\sum_{j=1}^{n_i}\sigma_i^j|y_i^j| \right| \\
&\leq \frac{4\gamma K}{M}\mathbb{E}\|\mathcal{W}\|_{**} + \frac{2K}{Mn}\mathbb{E}\left| \sum_{i=1}^{M}\sum_{j=1}^{n_i}\sigma_i^j|y_i^j| \right|.
\end{aligned}
\tag{48}
$$

Notice that

$$\mathbb{E}\left|\sum_{i=1}^{M}\sum_{j=1}^{n_i}\sigma_i^j|y_i^j|\right| \le b\mathbb{E}\left|\sum_{i=1}^{M}\sum_{j=1}^{n_i}\sigma_i^j\right| = b\mathbb{E}\left[\sqrt{\left(\sum_{i=1}^{M}\sum_{j=1}^{n_i}\sigma_i^j\right)^2}\right]$$

$$\le b\left[\mathbb{E}\sqrt{\left(\sum_{i=1}^{M}\sum_{j=1}^{n_i}\sigma_i^j\right)^2}\right] = b\sqrt{\sum_{i=1}^{M}n_i} = b\sqrt{N}.$$

(49)

Combining Eq.(46-49), we deduce that

$$\mathcal{R}(\hat{\mathcal{W}}) - \mathcal{R}(\mathcal{W}) \le \frac{4\gamma K}{M}\mathbb{E}[\|\mathcal{W}\|_{*^\star}] + \frac{2bK\sqrt{N}}{Mn} + a\sqrt{\frac{2log(2/\delta)}{Mn}},$$

(50)

with probability at least $1 - \frac{\delta}{2}$. The proof of this theorem is completed.

