# OpenReview forum: "A Swiss Army Knife for Heterogeneous Federated Learning: Flexible Coupling via Trace Norm"
_NeurIPS.cc/2024/Conference — NeurIPS 2024 poster_

### Official Review · Reviewer_zrPd · 2024-07-04

**Soundness:** 3
**Presentation:** 3
**Contribution:** 3
**Rating:** 7
**Confidence:** 4

**Summary:**

The authors present a new Swiss Army Knife-like approach that flexibly handles the heterogeneity challenges of a federated multi-task learning framework. The framework uniquely integrates tensor trace paradigms to handle cross-client data, model, and task heterogeneity. The title of the approach is interesting, but some problems remain.

**Strengths:**

1.The paper is easy to understand and comprehend, allowing even readers less familiar with the field to follow the authors' ideas.

2.The authors have made significant progress in addressing data heterogeneity, model heterogeneity, and task heterogeneity, and this ability to address heterogeneity in an integrated manner is a major highlight of this research.

3.The authors provide convergence guarantees and generalization bounds for the proposed FMTL framework in a non-convex setting, which further demonstrates the robustness and efficiency of FedSAK. These theoretical supports not only enhance the reliability of the method, but also demonstrate its potential in practical applications.

**Weaknesses:**

1.The authors consider the common assumptions that 1) the Lipschitz continuous gradient and 2) the variance is bounded. These assumptions are very stringent and the authors do not have a sufficient basis for these assumptions, adequate explanations, or empirical verification.

2.For a better understanding of the proposed technique, the authors may want to clarify some technical details. For example which layers of the model actually need to be shared in different heterogeneous setups.

3.The experimental results seem to show that the proposed algorithm has marginal gain compared to FedProto. At the same time, I would like to know why FedMTL has a faster convergence rate in Fig. 2.

4.The authors' model may significantly increase its computational cost as the size of the dataset increases and the network structure increases. Therefore the authors should go really such issues to focus on to enhance the generalization of the model.

**Questions:**

1.Can you provide more detailed information on how the cost of communication is calculated in Figure 5? The author's scale is too close to FedAvg to detect this baseline.

2.Whether the task heterogeneity proposed by the authors is a sub-case of model heterogeneity.

**Limitations:**

Please see weaknesses and questions

---

> ### Author Rebuttal · Authors · 2024-08-04
>
> Thank you for your valuable suggestions, your positive comments help us a lot. I will respond to your concerns next.
>
> > **Weaknesses 1: Reasonableness of assumptions**
>
> Thank you for your question, and I understand that these assumptions may be overly idealized in some cases, but Lipschitz continuous gradient, unbiased gradient, and bounded variance are standard assumptions for FL analysis [1-3]. Although these nonconvex assumptions are relatively strict, it remains an open question how to provide FL analysis without them.
>
> The  **Lipschitz continuous** gradient is a technical condition on the shape of the loss function and is the standard for nonconvex analysis. In contrast to previous assumptions on convex loss functions that are common in analyzing FL and stochastic gradient descent, the Lipschitz continuity assumption actually relaxes these conditions, allowing our analysis to cover a wider range of application scenarios.
>
> The  **bounded gradient** assumption allows our convergence analysis to better accommodate the heterogeneity of the data distribution, which is one of the core challenges of federated learning. In particular, the bounded gradient assumption is especially relevant in the case of certain activation functions (e.g., sigmoid functions) and bounded input features. Thus, this assumption not only provides an analytical basis in theory, but can also help cope with uneven data distribution in FL in practice.
> Our theory is based on the convergence results of the algorithm of [3], and in addition we also perform a verification convergence analysis in the experimental part as a proof that our algorithm is convergent.
>
> [1] On the Convergence of FedAvg on Non-IID Data. ICLR 2019.
>
> [2] Fedproto: Federated prototype learning across heterogeneous clients. AAAI 2022.
>
> [3] Heterogeneous federated learning with generalized global header. MM 2023.
>
> ***
>
> > **Weaknesses 2: Technical details of model sharing**
>
> Thank you for your valuable comments, and we apologize for the difficulty in understanding the shared layer structure due to the lack of detailed description in the paper. To address this issue, we provide a clearer description here and will make corresponding changes in the paper to enhance readability and reproducibility.
>
> In our approach, the local model is decoupled into a feature extraction part $\theta$ and a prediction head part $\varphi$.
>
> In the **data heterogeneity** scenario, we choose to share the whole model structure since it is the same for all the clients, as emphasized in line 286 of the paper.
>
> In contrast, in the **model heterogeneity** scenario, we share only the prediction header part, which is illustrated in line 298, i.e., the final fully connected layer shown in Table 4 of the paper.
>
> For the **task heterogeneity** scenario, we on the other hand share the feature extractor part, i.e., all the structures in the model except the final classification layer, which is described in line 309.
>
> ***
>
> > **Weaknesses 3: Description of experimental results**
>
> FedSAK shows advantages not only for data heterogeneous scenarios but also in **model and task heterogeneous** scenarios. From the results, there is no baseline comparable to FedSAK across all heterogeneous scenarios and datasets. We counted the results of the average growth of data heterogeneity:
>
> |                             | FedAvg | pFedMe | Ditto | FedU | FedProto | FedGH |
> | :-------------------------- | ------ | ------ | ----- | ---- | -------- | ----- |
> | Relative improvement in acc | 13.2   | 3.06   | 2.47  | 3.03 | 1.22     | 3.24  |
>
> Our improvements are valuable.
>
>  In addition, the effectiveness and flexibility of FedSAK is an important step forward compared to previous work that focused primarily on data heterogeneity. The regularization term of FedMTL differs from that of our design in that the former's regularization term, which primarily employs the L2 norm enables FedMTL to converge faster in the early stages of training.
>
> In contrast, our approach better captures inter-task property differences by stacking models from different tasks and computing trace paradigms to encourage these task models to be represented in a shared low-dimensional space. As a result, our approach may be more sensitive to the regularization term, e.g.,  **in Fig. 6**, we show the significant effect of the regularization parameter lambda on model convergence.
>
> ***
>
> > **Weaknesses 4 & Questions 1: Communications overhead**
>
> Q1: In the case of data heterogeneity (i.e., model homogeneity), our communication cost is the same as FedAvg, because we are uploading all models to the server. Whereas, in the case of model heterogeneity and task heterogeneity, we are sharing only some of the models, which will greatly reduce our communication cost.
>
> W4: It is due to the flexibility of FedSAK to choose the model sharing structure, which provides the possibility to flexibly cope with different model sizes and structures in large-scale federated learning environments. We provide a description of the computational complexity in **Appendix A.5.** In addition, due to the word limit, the experimental results in this section can be found in **Weaknesses 1 of our response to reviewer yu6g.**
>
> ***
>
> > **Questions 2: Task heterogeneity**
>
> In our study, although each task is associated with its unique classifier, these classifiers typically share the same model architecture. The task heterogeneity primarily manifests in the differences in classification objectives and data characteristics. Therefore, we do not classify this scenario under model heterogeneity. **For example**, in the context of drug classification, two clients may have tasks that involve binary classification for toxicity and activity, respectively. While both tasks use the same binary classifier architecture, their labels and objectives are entirely different, exemplifying task heterogeneity rather than model heterogeneity.

---

> > ### Comment · Reviewer_zrPd · 2024-08-08
> > **Response**
> >
> > Thanks. My issues has been addressed, so I raise the score to 7

---

> > > ### Author Response · Authors · 2024-08-08
> > >
> > > Dear reviewers,
> > >
> > > We thank you for scrutinizing the discussion and raising the score. We are glad we were able to address your concerns. If you have any further questions, please feel free to let us know. We are more than happy to answer them for you.

---

### Official Review · Reviewer_yu6g · 2024-07-10

**Soundness:** 4
**Presentation:** 3
**Contribution:** 3
**Rating:** 6
**Confidence:** 5

**Summary:**

This paper focuses on the issues of heterogeneous federated learning, including data heterogeneity, model heterogeneity, and task heterogeneity. To address these issues, the authors introduce a federated multi-task learning framework based on tensor trace norm.

**Strengths:**

1. Innovativeness: The authors focus on task heterogeneity scenarios, which sets this paper apart from other common federated learning studies. The ideas and topics presented by the authors are both intriguing and relevant.

2. Flexibility: FedSAK demonstrates strong adaptability to various forms of heterogeneity in federated environments, showcasing its generalization capabilities. Additionally, the framework enhances its adaptability through flexible upload and sharing structures.

3. Theoretical Analysis: This paper provides theoretical analysis as a guarantee.

4. Experiments: Evaluations on six real-world datasets show that FedSAK outperforms existing state-of-the-art models in terms of accuracy and efficiency, proving its effectiveness in handling heterogeneous federated learning scenarios.

5. Code: The author provides the code and is able to reproduce.

**Weaknesses:**

1. Computational complexity: As far as we know, the calculation of tensor trace norm is complex, so it is worth considering how to apply FedSAK to larger models.

2. Contribution: Although the author's innovation is commendable, the author did not clearly emphasize in the paper why the trace norm can make outstanding contributions. Can relevant ablation experiments be designed to further confirm the collinearity of the algorithm.

3. Convergence analysis: The author followed the common convergence analysis of federated algorithms. Can the author explain how the local convergence proposed by the author in Section 5.1 ensures consistency with achieving global optimization objectives?

**Questions:**

1. Does the author's stacking operation on the server result in a significant increase in algorithmic memory?

2. By which dimension are the authors stacking convolutions in model heterogeneous scenarios?

3. Does stacking in different dimensions affect the model? The authors should discuss more details of the model.

**Limitations:**

The author's limitations may be at a computing cost, but this is acceptable. It would have been more reasonable if this provided a short and clear section on the limitations.

---

> ### Author Rebuttal · Authors · 2024-08-04
>
> Thanks for your affirmative suggestions, which are of great support to us. Next, I will answer your questions one by one.
>
> > **Weaknesses 1: Computational complexity**
>
> The tensor trace norm is defined as the sum of matrix singular values, and its computational complexity on the server is
> $O(\min_k d_k^2·\prod_{i\ne k}^pd_i)$, where $d_i$ denotes the $i$-th dimension of the tensor.
>
> This may indeed increase as the size of the network or dataset increases. However compared to traditional methods that require uploading the entire model parameters e.g. FedAvg, FedProx, etc., our FedSAK method shows some flexibility in dealing with larger networks or datasets. We can **selectively share part of the model's structure**, which offers the possibility to flexibly cope with different model sizes and structures in large-scale federated learning environments. We provide a description of the computational complexity in **Appendix A.5**, which we tested using **Resnet18** (FedSAK only uploads the last FC):
>
> |            | FedAvg | pFedMe | Ditto | FedMTL | FedProto | FedGH | FedSAK |
> | :--------- | ------ | ------ | ----- | ------ | -------- | ----- | ------ |
> | ACC (%)        | 68.05  | 75.84  | 76.86 | 73.68  | 78.34    | 75.95 | 77.69  |
> | TIMES (s)  | 6729   | 57406  | 19366 | 11757  | 12076    | 7508  | 7118   |
> | MEMORY (G) | 1.75   | 2.63   | 3.42  | 2.58   | 1.75     | 1.71  | 0.918  |
>
> ***
>
> > **Weaknesses 2: Contribution**
>
> Thank you for recognizing our innovativeness.
>
> The core idea of federated multi-task learning is to improve learning across tasks by sharing information, where different tasks share certain features or structures. **The tensor trace norm** encourages models of different tasks to be represented in a shared low-dimensional space by constraining the low-rank nature of the weight matrices, thus effectively capturing the common information among tasks.
>
> For example, the parameters of multiple tasks can be viewed as different dimensions of a tensor, and the tensor trace paradigm automatically models inter-task dependencies by constraining the low-rank of this tensor and inducing the parameter matrices between different tasks to have a similar low-rank structure. Such dependencies help information sharing between tasks and improve the overall learning performance.
>
> Unlike common ablation experiments, in our approach, after removing the trace paradigm the model will degrade to a local model if it is not aggregated, and will degrade to a FedAvg if the model is simply weighted and aggregated. Therefore, we put this part of the special note in the **Appendix, see A.4 and Fig. 8.**
>
> ***
>
> > **Weaknesses 3: Convergence analysis**
>
> We provide guarantees for the global optimization in the convergence analysis through the derivation of **Assumption 3 and Lemma 2**.
>
> In the convergence analysis, **Assumption 1** shows that the local objective function is continuous and smooth, **Assumption 2** that the variance of the stochastic gradient over a batch of data is constrained by a constant, and **Assumption 3** that the difference between the parameters of the local shared layer and the updated parameters of the shared layer on the server side is bounded (here associated with global optimization).
> Based on the above assumptions, we show by **Lemma 1** that the loss of the local model for any client is bounded.
>
> **Lemma 2** further shows that after each round of communication, when the client replaces its local structure with the server's latest global shared layer, the loss of the local model is still bounded, thus helping to ensure the convergence of the model during training. Finally we obtain **Theorem 4 see Eq. 24** when we carry Lemma 1 into 2, which shows that the loss of the local model of any client is reduced in one round of communication with respect to the previous round, as a way to perform the convergence analysis.
>
> ***
>
> > **Questions 1: Memory problem**
>
> Thank you for your valuable input and inspiration. The fact that the memory overhead of our model is comparable to the base method FedAvg is due to the fact that we both need to upload the overall model to the server side. Our method just has the extra operation of stacking the models together and there is no additional memory constraint.
>
> ***
>
> > **Questions 2 & Questions 3: The stacked form problem**
>
> In our scenario, we adopt a client dimension stacking model-based approach for processing.
>
> Specifically, suppose we have $M$ clients, and the corresponding model weight matrix of each client is denoted as $\theta \in R^{d_{1} \times d_2}$ . After the training is completed, each client uploads its local model weights to the server. Then, the server stacks these model weights from different clients according to the client dimensions to form a new tensor $\Theta \in R^{d_{1} \times d_2  \times M}$. The third dimension of this tensor corresponds to the different clients, such that the model weights of all clients be integrated in a unified tensor structure. This stacking method can effectively centralize the model information of each client, which facilitates further global model update or analysis.
>
> It is worth noting that we are stacking parameters of the same size. In this way the correlation of different client models can be better explored. We have discussed the models for model heterogeneity in **Table 4**, and we will revise this section in detail and mention it in the main text.

---

> > ### Comment · Reviewer_yu6g · 2024-08-10
> >
> > The authors replied carefully and basically solved my concerns.

---

> > > ### Author Response · Authors · 2024-08-12
> > >
> > > Dear Reviewer.
> > >
> > > Thank you for your careful review of our discussion. We are happy to address your concerns. If you have any other questions, please feel free to let us know. We are more than happy to answer them.

---

### Official Review · Reviewer_kdUe · 2024-07-11

**Soundness:** 3
**Presentation:** 4
**Contribution:** 3
**Rating:** 7
**Confidence:** 4

**Summary:**

This paper introduces a federated learning method called FedSAK. Compared to existing methods, FedSAK is more flexible and can accommodate data heterogeneity, model heterogeneity, and task heterogeneity. To achieve knowledge transfer between client models in a heterogeneous environment, this method employs tensor trace norm regularization. The authors provide both theoretical and empirical evidence to demonstrate the effectiveness of this approach.

**Strengths:**

This paper is highly motivated and presents a clear and easy to understand methodology. The authors explain the methodology thoroughly and the logic is clear and easy to understand.
Experiments conducted on several different datasets demonstrate the effectiveness of the proposed methodology. The results consistently demonstrate superior performance compared to existing methods, highlighting the robustness and versatility of FedSAK.
Moreover, the method is theoretically sound and experimentally validated.

**Weaknesses:**

The authors should have provided more specific ablation experiments to assess the effectiveness of the key components of the method.
The authors should explain why such a division of the data was used in the common case of data heterogeneity and whether this is more realistic. For example, in Table 1, there are four combinations of M and S on CIFAR-10 and CIFAR-100, but only two on the first two datasets.
The reviewer wonders why the authors focalize with the task heterogeneity scenario, which can be seen in the paper actually task heterogeneity each task has different classifiers, whether this is also a model heterogeneity scenario. If yes why it needs to be divided separately, if not the authors should provide more explanation for this scenario.

**Questions:**

See above

---

> ### Author Rebuttal · Authors · 2024-08-04
>
> Thanks for your recognition and valuable suggestions, I will respond to your questions next.
>
>  ***
> >**Weaknesses 1: Ablation experiments**
>
> We appreciate your valuable comments. We understand the importance of ablation experiments in verifying the validity of a method. Due to specific design choices, we provide the results of the ablation experiments in **Appendix A.4, see Fig. 8**.
>
> In the scenario of data heterogeneity, the models uploaded by FedSAK are isomorphic, which implies that FedSAK will degrade to the standard Federated Averaging (FedAvg) algorithm if weighted aggregation is performed only on the server side, as shown in the results in **Table 1.** This effectively amounts to an implicit ablation experiment, demonstrating the performance degradation that will occur when key innovations in our approach are removed.
>
> For the model-heterogeneous and task-heterogeneous scenarios, what we upload to the share is a portion of the local model. If nothing is done to this part of the shared model, then our method will degrade to a locally trained model only, i.e., the Local model in **Tables 2** and **Table 3**. On the other hand, if a simple weighted aggregation is used, our method will degenerate into the FedAvg-c model in **Table 3**. These degradation scenarios can actually be considered as a form of ablation experiment, demonstrating the importance of the individual components of our method.
>
> We show the results of the ablation experiments based on the above analysis as follows, which are described in detail in Appendix A.4 of this paper:
>
> |                                   | Local | FedAvg | FedSAK    |
> | :-------------------------------- | ----- | ------ | --------- |
> | Data heterogeneity (CIFAR10)      | 68.72 | 66.55  | **76.47** |
> | Model heterogeneity (PACS)        | 59.13 | 60.98  | **68.05** |
> | Task heterogeneity (Adience Face) | 72.26 | 74.73  | **76.03** |
>
> Thus, in **Appendix A.4**, we provide the results of the ablation experiments described above. However, our results do demonstrate the degradation of performance when key features of our method are removed or simplified. We will make this clearer in the revision of the paper so that the reader can better understand the effectiveness of our method and the role of the individual components.
>
> ***
> > **Weaknesses 2: Explanation of dataset division**
>
> Thank you for your question, we use the dataset partitioning method commonly used by our predecessors in federated multitask learning, **see [1, 2]** for references. A detailed explanation of the dataset can be found in **Appendix B.1**. As we mentioned in A.1 the dataset HumA contains the behavioral actions of 30 individuals, each of which is fixed to correspond to 6 action categories, so we keep the number of client categories constant.
>
> Similarly, the MNIST dataset is an federated multi-task learning dataset divided based on previous experience. In contrast, the CIFAR-10 and CIFAR-100 datasets are more complex than MNIST, with more categories and more complex images. By using different number of clients and number of labeled categories, the robustness of the model in dealing with complex data distributions can be better evaluated.
>
> [1] Smith V, Chiang C K, Sanjabi M, et al. Federated multi-task learning[J]. Advances in neural information processing systems, 2017, 30.
>
> [2] Dinh C T, Vu T T, Tran N H, et al. Fedu: A unified framework for federated multi-task learning with laplacian regularization[J]. arXiv preprint arXiv:2102.07148, 2021, 400.
>
> ***
> > **Weaknesses 3: Differences between task heterogeneity and model heterogeneity**
>
> Thank you for your question, first of all task heterogeneity scenarios are very common in real world applications, where different tasks usually have different feature distributions and classification needs. **Task heterogeneity emphasizes the fact that each task has its own unique features and goals**, which often require independent classifiers to handle. Therefore, although each task uses a different classifier, we focus primarily on the heterogeneity of the task itself rather than the heterogeneity of the model.
>
> In our work, task heterogeneity and model heterogeneity are indeed somewhat related, but they are **not the same concept**. Task heterogeneity refers to differences in data distribution and objectives between tasks, while model heterogeneity refers to the use of different model architectures for different tasks.
>
> In our study, although each task has its unique classifiers, these classifiers do not necessarily use completely different model architectures, and thus we did not categorize them as model heterogeneity scenarios. For example, when doing drug classification, the two clients' tasks were binary classification tasks of determining the presence or absence of toxicity and the presence or absence of activity, respectively. Their classifiers are both binary classification tasks, but the task labels are completely different.

---

> > ### Comment · Reviewer_kdUe · 2024-08-11
> >
> > I am satisfied with your responses. Thus, I improve my score slightly.

---

> > > ### Author Response · Authors · 2024-08-12
> > >
> > > Dear Reviewer,
> > >
> > > Thank you for scrutinizing our discussion, and we are pleased to have been able to address your concerns. Thank you for improving your score for our paper. If you have any other questions, please feel free to let us know. We are more than happy to answer them.

---

### Author Rebuttal · Authors · 2024-08-05

We thank the reviewers for providing detailed and thoughtful feedback on our submission. We thank the reviewers for appreciating our innovation and value to a large extent while suggesting improvements. We have addressed reviewer comments and questions in individual responses to each reviewer and in the accompanying pdf file. If any questions were not answered or our responses were unclear, we would appreciate the opportunity to engage further with our reviewers.

Briefly, the main points of our response are as follows:

**1. Regarding the ablation experiments**: reviewers kdUe and yu6g point out that more specific ablation experiments should be provided. We thank the reviewers for their attention to this issue, and due to our specific experimental setup, our experimental results actually provide an implicit ablation. We show the experimental results in our kdUe response.

**2. Regarding the computational complexity of the model:** reviewers yu6g and zrPd are concerned about the computational complexity of our model. We recognize that the computational complexity of the tensor trace paradigm increases with the size of forgetting. However, since our model is a model structure that can flexibly choose shared partial uploads, we can largely environment the stress caused by the increase in model size, which can be reacted to in our experiments, and draw attention to the brief experimental results in Appendix A.5.

**3. Explanation on task heterogeneity:** Both reviewers kdUe and zrPd raised questions about our proposed task heterogeneity scenario. We regret that we did not describe this task scenario carefully enough in the paper. We will revise it in a subsequent version.

We thank you again for your time and effort in reviewing this submission, and we are confident that these comments will enhance the clarity and motivation of our manuscript.

---

### Decision · Program_Chairs · 2024-09-25

**Decision:**

Accept (poster)

**Comment:**

The authors present a new Swiss Army Knife-like approach that flexibly handles the heterogeneity challenges of a federated multi-task learning framework. The proposed framework is novel and interesting with both theoretical and empirical evidence to demonstrate its effectiveness. Some concerns in the initial reviews such as computational cost, ablation experiments and convergence analysis have been successfully addressed by the authors' rebuttal.